# FOSTERING VIDEO REASONING VIA NEXT-EVENT PREDICTION

**Haonan Wang**[*][N]  **Hongfu Liu**[*][N]  **Xiangyan Liu**[N]
**Chao Du**[S]  **Kenji Kawaguchi**[N]  **Ye Wang**[N]  **Tianyu Pang**[†][S]
[N]National University of Singapore   [S]Sea AI Lab, Singapore

## ABSTRACT

Next-token prediction serves as the foundational learning task that enables reasoning in LLMs. But what should the learning task be when aiming to equip MLLMs with *temporal reasoning* capabilities over video inputs? Existing tasks such as video captioning primarily promote modality alignment, while video question answering typically relies on annotations from humans or much stronger MLLMs. To address this gap, we propose **next-event prediction (NEP)**, a learning task that harnesses future video segments as a rich, self-supervised signal to foster temporal reasoning. We segment each video into **past** and **future** frames: the MLLM takes the past frames as input and predicts events in the future, thereby encouraging the model to reason temporally in order to complete the task. To study this learning task, we curate **V1-33K**, a dataset comprising 33,000 automatically extracted videos spanning diverse real-world scenarios. Using the same videos, we further explore a range of video instruction-tuning task data to provide controlled comparisons and isolate the effect of NEP. To evaluate progress, we introduce **FutureBench** to assess coherence in predicting unseen future events. Experiments validate that NEP offers a scalable and effective training task for fostering temporal reasoning in MLLMs.

## 1 INTRODUCTION

Recent progress in multimodal large language models (MLLMs) has significantly advanced video understanding capabilities (Hurst et al., 2024; Team, 2025). Video instruction tuning typically involves *learning tasks* such as video question answering, captioning, and grounding, etc., which mainly emphasize visual perception skills such as object identification, event recognition, timestamp inference, and factual recall based on observed video frames (Bai et al., 2025; Li et al., 2024c; Lin et al., 2023; Zhang et al., 2024b). While these tasks facilitate cross-modal alignment—an essential step in integrating visual encoders with language models (Liu et al., 2023)—they often neglect the *temporal* dimension that distinguishes videos from static images. For instance, video question answering frequently relies on key frames (Cores et al., 2025), and video captioning tends to map frames to text in a frame-by-frame manner, limiting the model's ability to understand dynamic event progression. Moreover, tasks like question answering and timestamp grounding typically require annotations by humans, raising scalability challenges. This leads to a natural question:

> *What learning task should be employed to effectively equip MLLMs with temporal reasoning capabilities over video inputs?*

To bridge this gap, we propose **Next-Event Prediction** (NEP), a learning task explicitly designed to foster temporal reasoning in MLLMs. Instead of providing the entire video as input, NEP splits each video into *past* and *future* frames. Given only the past frames, the model predicts captions that describe the future segment, as illustrated in Figure 1. NEP encourages MLLMs to reason beyond the visible scene, inferring causes, effects, and likely outcomes. The NEP naturally drives the model to integrate visual perception from the visual encoder with commonsense knowledge in the LLM.

To systematically study the effectiveness of NEP as a learning task, we construct **V1-33K**, a large-scale dataset comprising approximately 33,000 automatically curated videos. Each instance consists of an

---

[*]Equal contribution. Work done during Haonan Wang and Hongfu Liu's internships at Sea AI Lab.
[†]Correspondence to Tianyu Pang.

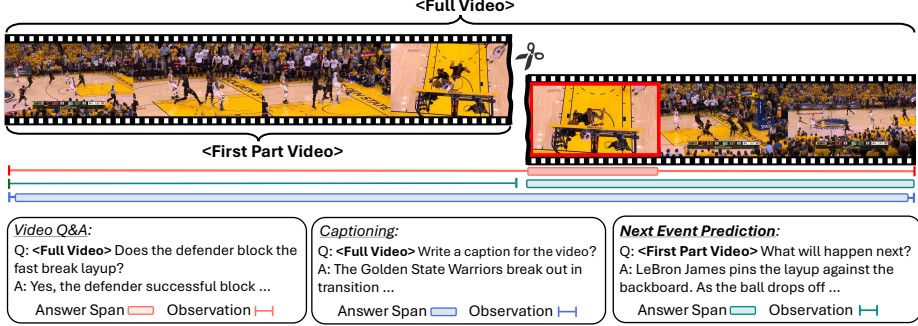

Figure 1: **Comparison of Video Instruction Tuning tasks.** (1) **Video Q&A**: Extracting answers from a single key frame; (2) **Captioning**: Summarizing from frame-by-frame visual perception of observed videos; (3) **Next-Event Prediction**: Predicting the summary of future frames by visual perception of observed past frames and temporal reasoning with commonsense knowledge. As the example in the given first part video, after a defensive stop, the team may push fast in transition (knowledge)—but with under two minutes left in the fourth quarter (visual facts), a coach might call a timeout, or the players may slow the tempo to ensure careful execution.

observable (past) video segment paired with a textual caption describing its subsequent continuation, which serves as the supervision target. The V1-33K is derived from the video data source used by previous works for video-instruction tuning (Zhang et al., 2024b; Li et al., 2024a), and thus naturally includes data suitable for other learning tasks, such as captioning and question answering. Accordingly, we train the same models on the same data but different learning tasks, enabling a controlled study of NEP's effect while holding other factors (e.g., data source bias and data quality) constant.

Moreover, we further explore the NEP task with different training strategies, including standard supervised fine-tuning (SFT) (Ouyang et al., 2022), critique fine-tuning (CFT) (Wang et al., 2025), teacher model distillation (Distill) (Ho et al., 2022), and R1-style reinforcement training (Shao et al., 2024). To rigorously assess the temporal reasoning capabilities, we introduce **FutureBench**, a comprehensive benchmark evaluating logical coherence and causal consistency in predicting unseen future events. FutureBench challenges models to perform multi-hop temporal reasoning by generating plausible event sequences that bridge observed video segments and specified future outcomes. Empirically, across three prior temporal benchmarks – TempCompass (Liu et al., 2024b), TemporalBench (Cai et al., 2024), and SEED-Bench-R1 (Chen et al., 2025) and our FutureBench, NEP as a learning task significantly enhances MLLMs' temporal understanding and reasoning. Moreover, on general-ability suites – VideoMME (Fu et al., 2024), MVBench (Li et al., 2024d), LongVideoBench (Wu et al., 2024), models trained solely on NEP preserve their performance on conventional video tasks.

## 2 RELATED WORK

**Video Instruction-Tuning of MLLMs.** The fusion of vision and language in large models has advanced rapidly from image-focused models like CLIP (Radford et al., 2021) and LLaVA (Liu et al., 2023) to recent video-language models that interpret dynamic visual content leveraging the advanced ability of LLMs (Liang et al., 2024; Tang et al., 2023). Early approaches adapted image-based techniques by fine-tuning LLMs with an extended visual encoder on video frames for observational tasks, such as captioning and question answering; this process is also known as video instruction tuning. Models such as Video-LLaVA (Lin et al., 2023), LLaVA-NeXT series (Li et al., 2024a;c; Zhang et al., 2024b) and Qwen-VL series (Bai et al., 2023; 2025) fine-tune large language models with video-frame inputs, enabling open-ended video description and Q&A. These MLLMs demonstrate strong performance on tasks like captioning and dialogue about videos. However, their training data and objectives are predominantly observational, describing or explaining visible content, rather than predictive. Our work differs by introducing a predictive objective, next event prediction, to explicitly train the model's temporal reasoning abilities. This aligns with the goal of modeling world dynamics, extending beyond static understanding of frames to reasoning about how scenes evolve over time.

**Future Prediction in Computer Vision.** Anticipating the future has been studied in computer vision in various forms. Action prediction methods train video encoders by predicting the next action or action label from representations encoding past video frames (Beedu et al., 2024; Liu et al., 2024a; Stergiou & Damen, 2023; Tran et al., 2021; Gammulle et al., 2019; Lan et al., 2014; Miech

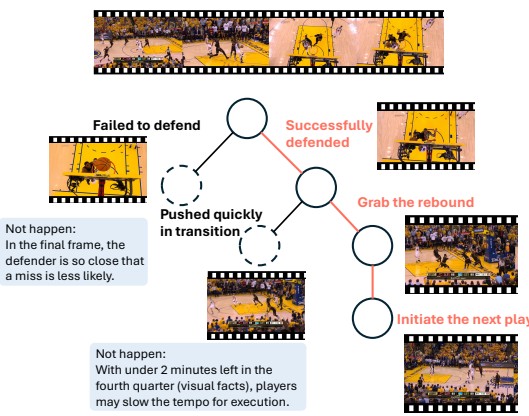

Figure 2: **Reasoning structure underlying NEP**. Each node is a potential event or action derived from visual cues, branching into alternative scenarios such as failing to defend or being pushed in transition. The red line highlights actual event sequence observed in the video. Comments provide reasoning for less likely scenarios.

Figure 3: **Distribution of data source and video length in V1-33K**. The inner circle illustrates the distribution of data sources. The outer circle further segments each source according to video length categories. Only length categories comprising more than 4% of the dataset are labeled explicitly in the outer circle.

et al., 2019). Similarly, future-frame prediction and motion forecasting train video encoders via self-supervised objectives, e.g., predicting representations of future frames (Ranzato et al., 2014; Vondrick et al., 2016; Liu et al., 2025; Assran et al., 2025). Hosseinzadeh & Wang (2021) not only train a video encoder, but also separately train a translator that converts these predicted features about future into textual descriptions of the anticipated video content. These works typically define the "*future*" at short horizons, e.g., the next frame or next action; and optimize low-level objectives, e.g., representation learning, which has proven effective for video encoder pretraining.

Our work is distinct in that it focuses on high-level, semantic future-event prediction expressed in natural language. This task requires MLLMs to perceive temporally visual information via the vision encoder and then integrate it with pre-trained commonsense knowledge in the LLM to generate open-ended natural language responses. The goal is different, therefore, rather than improving a video encoder, as previous methods attempted, we aim to improve the cross-modal projector and LLM of the MLLM while freezing visual encoder parameters. Due to the limited space, we defer the discussion of Related Work to Appendix A.

## 3 NEXT-EVENT PREDICTION

Our work focuses on the study of NEP as a learning task for MLLMs. In this section, We first formalize the NEP task, then describe the constructed dataset for studying it, and finally detail training strategies (*e.g.*, SFT, CFT, and Distillation) used in combination with NEP.

### 3.1 FORMULATION

We formulate NEP in a video as a sequence-to-sequence language modeling problem conditioned on video frames. Supposing $V = [v_1, v_2, \ldots, v_T]$ represents a sequence of video frames (or clips), a cut-off time $t < T$ is chosen to split the full video into an observed part $V_{\leq t} = [v_1, \ldots, v_t]$ (past frames) and a future part $V_{>t} = [v_{t+1}, \ldots, v_T]$ (future frames). The goal is to train an MLLM that takes $V_{\leq t}$ as input and generates a textual $Y$ of events in $V_{>t}$. In practice, for SFT training strategy, $Y$ can be simply represented by the token sequence of captions in future frames. For the details of other training strategies, like CFT and distillation, we detail in Section 3.3.

**Why NEP can be effective.** Because the target text describes events not visible in $V_{\leq t}$, NEP pushes the model beyond perception (e.g., object detection or action recognition) to prediction. It requires MLLMs to infer object interactions and dynamics, project them forward in time, and maintain narrative coherence. To predict plausible next events, MLLMs must integrate two information sources: (i) perceived visual evidence from the visual encoder and (ii) commonsense world knowledge, such as physics, social norms, and typical human behavior from the LLM, as reasoning drivers. NEP

training therefore combines perception with knowledge-guided causal inference, resulting in improved temporal understanding and reasoning, which are required for complex video understanding.

## 3.2 DATASET CONSTRUCTION FOR STUDYING NEP

To systematically study NEP, we build **V1-33K**, a dataset produced by a simple yet effective pipeline that converts videos into training instances for NEP.

**Pipeline overview.** Given a video data source, we extract a *past* video segment (model input) and a *future* video segment together with text supervision signal (model target) via the following procedure.

1. ***(Optional) Caption Analysis.*** An LLM parses the video caption to identify distinct scenes and a causal pivot (the point where "what has happened" transitions into "what will happen"). This yields a *text split point*.

2. ***Grounding and Segmentation.*** We ground the *text split point* to the video timeline using an MLLM, obtaining a timestamp $t$. The video is cut at $t$ into **past** and **future** parts, and the caption is split accordingly into a *past-caption* and a *future-caption* (target).

3. ***(Optional) Reasoning & Critique.*** The past caption is processed by a *text* reasoning model to generate predictions and reasoning traces, which are then critiqued by another LLM.

Note, both *Caption Analysis* and *Reasoning & Critique* are optional. Caption analysis aims to improve the challenge of NEP by suggesting a split point in text format to avoid easy and less meaningful next-event prediction. If analysis is skipped, we fallback to sampling a mid-sentence pivot from the caption as the *text split point* and ground it and then split video in the same way. The reasoning and critique step aims to rollout reasoning traces and critiques data for critique fine-tuning and distillation in our ablation study of different training strategies. While both *Caption Analysis* and *Reasoning & Critique* are technically optional—the pipeline can still produce future prediction data without them. We enable them when constructing the V1-33K because they improve the quality and challenge level of the constructed data. We leave more details in Appendix B.

**NEP yields auto-labeled self-supervision signals.** NEP leverages signals already present in the data. The only prerequisite of the signal construction (*Grounding and Segmentation*) is timestamp grounding ability, aligning the caption split with the corresponding video cut. Following the common training recipe (Bai et al., 2025; Chen et al., 2024b; Zhang et al., 2024b), the model undergoes CLIP pre-training (for visual encoder), vision–language alignment (modality alignment), and then end-to-end fine-tuning (advanced abilities). NEP asks an MLLM to perceive visual information and then integrate it with the LLM module's commonsense knowledge to perform reasoning and predict future events. Considering that NEP requires MLLMs to perceive visual information and then integrate it with the LLM module's commonsense knowledge for reasoning and future prediction, it is reasonable to take NEP as the learning task in the late fine-tuning stage for unlocking advanced temporal ability, where the basic timestamp referencing/grounding ability has already been learned in an early fine-tuning stage. With this in place, the trained MLLM can reliably perform timestamp grounding, yielding auto-labeled self-supervision. Even considering the caption analysis, the LLM of MLLM is sufficient to complete this lightweight task.

**Collection and coverage of V1-33K.** Following this pipeline, we process thousands of videos from diverse sources (e.g., YouTube, YouCook2, NextQA, Charades, ActivityNet), containing 33,000 *past–future* pairs. The dataset spans rich scenarios—physical events (spills, collisions, object interactions), human interactions (arguments leading to reactions, pranks leading to surprises), and sports (setups leading to goals or failures). Source and duration statistics are summarized in Fig. 3.

**Data quality and source bias.** The quality of data has a strong influence on the performance of foundation models. In our work, to study the effect of NEP as a learning-task in comparison to others, we intentionally use the same video data source as other common instruction tuning tasks such as video captioning and QA. This allows training the same MLLMs on the same videos but with different tasks (captioning/QA vs. NEP), providing a controlled comparison that holds video quality and source bias fixed. The quality of NEP data is also determined by its textual supervision. Given the past video segment, plausible continuations vary widely, resulting in varying difficulty of next-event prediction—some trivial, some extremely challenging, others reasonably for the currently trained MLLMs. This mirrors next-token prediction in language, where the difficulty of the next token also varies; its success for foundation models comes from its ease of scaling up. We follow the

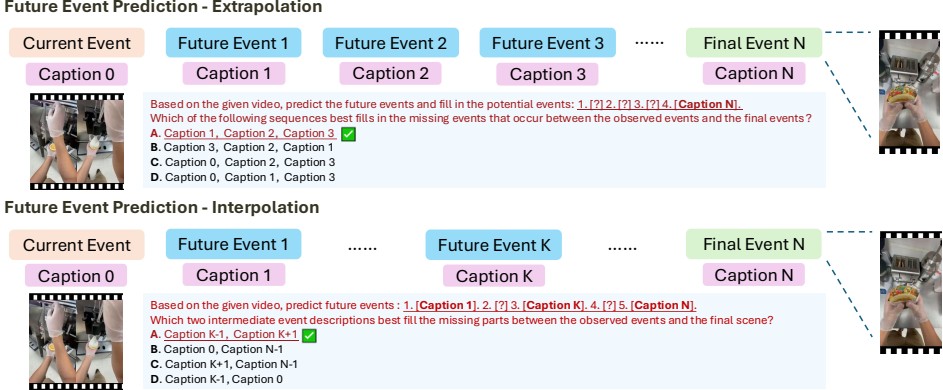

Figure 4: **Task demonstration of FutureBench**. This figure presents two paradigms for future event prediction: Extrapolation and Interpolation. In the **Extrapolation task (Top)**, the model observes the initial video (Current Event) and is required to sequentially predict a series of future events (Caption 1 → Caption 2 → Caption 3 → ...) leading up to the final event (Caption N). In the **Interpolation task (Bottom)**, the model observes the initial video (Current Event) and is provided with the first future event (Caption 1), an anchor future event (Caption K), and the final event (Caption N) and must infer the most plausible intermediate events that bridge the temporal gap. Distractors involve Caption 0 of the current event to require the model to understand the given video. Questions and answer options above are simplified for clarity and brevity.

same principle: our NEP data come from a simple, low-cost pipeline that automatically generates supervision. Our method is cheaper than video question answering. Note, even for humans at the PhD level, asking constructive questions is hard. For our data, we use the caption analysis step to cheaply improve target quality and rely on scalable automatic generation and accept some variation in difficulty. Despite the mixed quality of the data, our experimental results indicate the promise of using NEP as the learning task. And we expect NEP to benefit even more as data quality improves.

## 3.3 VIDEO INSTRUCTION-TUNING STRATEGIES

Besides the comparison in the task formulation, we further investigate four video instruction-tuning strategies on the NEP task. Each training strategy leverages specific annotations and structures from the V1-33K data pipeline, ranging from ground-truth next-event descriptions to critique and reasoning traces. We consider the encoder-decoder architecture model akin to recent MLLMs, LLaVA (Liu et al., 2023), where a vision encoder $E$ processes the video frames and produces a sequence of visual embeddings, and a language decoder $D$ attends to these embeddings to generate text. Specifically, for each input video $V_{\leq t}$, $E$ extracts frame features, and these features are fed into $D$ through a cross-attention mechanism. The decoder is then prompted to output the next-event description. During training, we supervise $D$ to match the ground truth next-event description using a standard language modeling loss, cross-entropy over the next token. We explore four distinct Video Instruction-Tuning strategies: supervised fine-tuning (SFT), critique fine-tuning (CFT), distillation tuning (Distill), and mix tuning (Mix), leveraging ground-truth video captions, critiques from GPT, and structured reasoning traces from DeepSeek. Details of tuning strategies are provided in Appendix D.

## 4 FUTUREBENCH

To advance the evaluation of MLLMs in temporal reasoning—specifically in forecasting future events from observed video—we introduce **FutureBench**, a benchmark designed to assess models' ability to infer plausible event progressions leading to a specified outcome. This task demands both strong visual perception and commonsense reasoning. Unlike prior video Q&A benchmarks, which focus on answer extraction from visible frames (Chen et al., 2025; Xiao et al., 2021), FutureBench emphasizes temporal-causal reasoning toward achieving unobserved future goals.

We formalize the evaluation in a multiple-choice question-answering format. Each video in FutureBench is paired with a clearly defined task end goal or event outcome, termed an "*end anchor*", which is derived from the final state of the full video. This design reflects the principle that real-world narratives typically take goal-driven paths, and it serves to limit the search space for potential future events. Given the anchor, the model must reason both *forwards* and *backwards* to deduce the plausible intermediate steps or events that lead to a specified outcome.

### 4.1 TASK SETTINGS: MULTI-HOP PREDICTION

A defining characteristic of FutureBench is its structured division into tasks with varying logical-hop distances, that is, the number of inferential steps or missing events the MLLM must predict. This design enables a comprehensive evaluation of both in-distribution performance on single-hop (1-hop) reasoning tasks and out-of-distribution generalization to more complex multi-hop reasoning involving extended event sequences. Accordingly, FutureBench is organized into two primary subtasks:

**Future Event Prediction—Extrapolation.** The extrapolation requires the model to predict a sequence of future events that logically connect the initial observed scenes to a specified final outcome. The task difficulty is controlled by varying the number of missing events, ranging from one to three:

- **1-Hop**: The model predicts a single future event that directly links the observed scenes to the final one. This corresponds to a standard NEP.
- **2-Hop**: The model infers a sequence of two consecutive future events, requiring a short chain reasoning process that sequentially connects the observed scenes to the final event.
- **3-Hop**: The model predicts three consecutive future events, significantly increasing task complexity by necessitating deeper causal reasoning across a longer temporal span.

**Future Event Prediction—Interpolation.** The interpolation subtask introduces a complementary challenge wherein the model must infer multiple non-consecutive future events, given a set of partially observed scenes that include intermediate anchor events. Rather than constructing a continuous sequence – as in extrapolation – this task demands the model interpolate across disjoint glimpses of future events. It emphasizes reasoning over causal continuity and temporal coherence amid fragmentary observation, as illustrated in Figure 4.

### 4.2 DETAILS OF QUESTION-ANSWER PAIR GENERATION

Designing high-quality questions and answer choices for FutureBench presents a non-trivial challenge, as it demands capturing the nuanced temporal logic embedded in each narrative. To scale the generation of QA pairs, we adopt a LLM-based generation pipeline. We employ GPT-4 (text-only mode) to generate QA pairs from detailed video annotations. Each video is accompanied by rich textual metadata, including a synopsis, segment-level scene descriptions, a specification of the observed scenes (i.e., the initial context), and a description of the final scene (i.e., the target outcome). We then prompt GPT-4 using a structured template designed to emulate a human question-setter. The prompt instructs GPT-4 to formulate a question that probes for the missing future events and to generate a correct answer along with several plausible yet incorrect distractors. **To ensure that the question requires genuine reasoning**, the prompt explicitly references the need to achieve a final outcome and is carefully crafted to prevent shortcut solutions – for examples, by avoiding lexical overlap between the correct answer and question, or easily dismissible distractors. Additionally, the distractor choices are constructed to be commonsense-plausible within the thematic context of the video but logically inconsistent with the outcome trajectory, thereby increasing task difficulty. An illustrative example of this process is shown in Figure 4, and the full prompt used for GPT-4 is provided in Appendix E. As a result, FutureBench comprises a total 1056 carefully curated QA pairs spanning both extrapolation and interpolation subtasks. To assess the benchmark's quality and highlight both visual perception and temporal reasoning, we evaluate a strong reasoning model, o4-mini, on the text-only version of questions, excluding any visual input. The model achieves an accuracy of 32.0%, suggesting that **even advanced reasoning capabilities alone are insufficient for consistently solving the tasks**. This finding reinforces the critical role of visual perception in solving future event prediction in FutureBench. More details regarding dataset distribution and the discussion of human-in-the-loop quality review in B.3.

## 5 EXPERIMENT

### 5.1 COMPARISON WITH OTHER VIDEO INSTRUCTION TUNING TASKS

To investigate the effectiveness of NEP as a learning task, we fine-tune Qwen2.5-VL-7B-Instruct on NEP and compare its performance against models trained on three prior instruction tuning tasks: captioning, multi-choice question answering (**MCQA**), and open-ended question answering (**OEQA**). For fairness, all models are trained on a dataset of equal size using 3K samples. For the captioning, MCQA and OEQA, we use the data constructed by LLaVA-Video-178K (Zhang et al., 2023).

To comprehensively evaluate model performance, we consider two groups of benchmarks. First, we assess general video understanding on three widely-used benchmarks that are not specifically designed

Table 1: **Performance comparison across different video instruction tuning tasks on Qwen2.5-VL-7B-Instruct.** G-Avg. and T-Avg. represent the average performances of all general and temporal benchmarks, respectively. Instruct represents the original performances without additional training.

| Task | General Benchmark | | | | Temporal Benchmark | | | | |
|------|-----------------|-----|-------|--------|------|------|-------|------|--------|
| | VMME(w/o sub) | MVB | LVB(val) | G-Avg. | TB | TC | SB-R1 | FB | T-Avg. |
| Instruct | 59.8 | 65.3 | 55.9 | 60.3 | 35.4 | 73.8 | 37.1 | 52.6 | 49.7 |
| Full Observed Video | | | | | | | | | |
| Captioning | **60.6** | 66.2 | 53.2 | 60.0 | 37.0 | 72.2 | 33.6 | 55.8 | 49.7 |
| MCQA | 57.4 | 65.2 | 53.0 | 58.5 | 32.1 | 65.5 | 33.0 | 60.3 | 47.7 |
| OEQA | 59.8 | **66.8** | 54.6 | 60.4 | 36.6 | 74.0 | 35.4 | 58.8 | 51.2 |
| Partially Observed Video | | | | | | | | | |
| NEP | 60.0 | 66.5 | **56.3** | **60.9** | **38.6** | **74.7** | **39.5** | **61.3** | **53.5** |

to test temporal reasoning: VideoMME(w/o sub) (**VMME**) (Fu et al., 2024), MVBench (**MVB**) (Li et al., 2024d), and LongVideoBench(val)(**LVB**) (Wu et al., 2024). Second, to examine temporal understanding and reasoning capabilities, we evaluate on four temporally-focused benchmarks: TemporalBench (**TB**) (Cai et al., 2024), TempCompass (**TC**) (Liu et al., 2024b), SeedBench-R1 (**SB-R1**) (Chen et al., 2025), and our proposed FutureBench (**FB**). These benchmarks challenge models to make complex temporal understanding and reasoning. For all evaluations, we use 32 frames from the video as the input by default. Detailed training and evaluation descriptions can be found in Appendix B.2.

**Next-event prediction enhances temporal reasoning without sacrificing general video understanding.** As shown in Table 1, models trained on the NEP task with partially observed video demonstrate substantial improvements on temporal benchmarks compared to those trained on Captioning, MCQA, and OEQA tasks with the full observed video. Notably, NEP-trained models also maintain competitive performance on general benchmarks, underscoring the superiority and compatibility of the NEP task. These findings suggest that NEP not only strengthens a model's ability to reason over temporal sequences but does so without compromising its overall comprehension abilities. NEP serves as an effective learning signal that promotes both visual perception and temporal reasoning with minimal trade-offs in general performance.

**Deductive reasoning via next-event prediction yields greater improvements on temporal benchmarks compared to inductive (video Q&A) and abductive (previous-event prediction) reasoning.** Video Q&A, next-event prediction, and previous-event prediction can be deemed three classical forms of logical reasoning—induction, deduction, and abduction, respectively (Douven, 2021; Cheng et al., 2024). An illustration is provided in Figure 17 in the appendix. To study the relative efficacy of these reasoning types, we fine-tune the Qwen2.5-VL-7B-Instruct model using the same training set of 3K samples, modifying only the task formulation to align with each reasoning. The results presented in Table 5 indicate that the deduction task, next event prediction, yields significantly greater performance on temporal benchmarks compared to induction and abduction tasks. In contrast to induction and abduction, deduction often involves the deliberate application of abstract logical principles. Such reasoning tends to be more cognitively demanding and typically necessitates targeted learning and structured practice (Goswami, 2010; Behfar & Okhuysen, 2018).

## 5.2 COMPARISON OF INSTRUCTION TUNING STRATEGIES

To further explore effective strategies for training on the NEP task, we compare four instruction tuning approaches introduced in Section 3.3: supervised fine-tuning (SFT), critique fine-tuning (CFT), distillation (Distill), and mix tuning (Mix). We conduct experiments on both Qwen2.5-VL-3B-Instruct and Qwen2.5-VL-7B-Instruct, evaluating each strategy across general and temporal video benchmarks. Additionally, we study the impact of training set size by scaling SFT and Distill from 1K to 25K samples, and CFT and Mix from 1K to 10K samples. For the Mix setting, we fix a total tuning budget and then combine examples from different supervision types in equal proportions (e.g., an equal number of SFT-, Distill-, and CFT-style NEP instances under the same total number of samples). During training, mini-batches are sampled from this unified mixed pool.

**SFT serves as a simple but effective strategy for NEP training.** As shown in Table 2, simple SFT yields substantial gains on temporal benchmarks, demonstrating its efficacy for NEP. While CFT and Distill also contribute notable improvements, they rely on additional annotations or feedback from auxiliary LLMs, making them less efficient in comparison to SFT. Importantly, Mix strategy achieves

Table 2: **Performance comparison of different instruction tuning strategies**. G-Avg. and T-Avg. represent the average performances of all general and temporal benchmarks, respectively. Instruct represents the original performances without additional training.

| Models | General Benchmark | | | | Temporal Benchmark | | | | |
|---|---|---|---|---|---|---|---|---|---|
| | VMME(w/o sub) | MVB | LVB_val | G-Avg. | TB | TC | SB-R1 | FB | T-Avg. |
| Qwen2.5-VL-3B-Instruct | | | | | | | | | |
| Instruct | 55.7 | 63.8 | 52.2 | 57.2 | 30.8 | **69.3** | 33.2 | 49.9 | 45.8 |
| SFT | 55.8 | 62.8 | 50.4 | 56.3 | 34.3 | 61.5 | **35.7** | **61.1** | 48.2 |
| CFT | 55.6 | 63.1 | 50.9 | 56.5 | 32.6 | 68.5 | 34.6 | 50.1 | 46.5 |
| Distill | 56.2 | 64.5 | **53.5** | **58.1** | 33.9 | 69.1 | 33.6 | 57.2 | 48.4 |
| Mix | **56.6** | **64.6** | 52.4 | 57.9 | **34.8** | 66.5 | **35.7** | 56.9 | **48.5** |
| Qwen2.5-VL-7B-Instruct | | | | | | | | | |
| Instruct | 59.8 | 65.3 | 55.9 | 60.3 | 35.4 | 73.8 | 37.1 | 52.6 | 49.7 |
| SFT | 59.2 | 66.5 | 53.4 | 59.7 | **39.9** | 69.9 | 39.1 | 61.3 | 52.6 |
| CFT | 58.9 | 65.3 | 54.2 | 59.5 | 35.2 | 74.1 | **39.8** | 55.8 | 51.2 |
| Distill | **60.6** | **66.7** | **56.3** | **61.2** | 35.9 | **75.1** | 37.0 | 59.5 | 51.9 |
| Mix | 59.6 | 66.4 | 53.7 | 59.9 | 38.2 | 72.9 | 38.5 | **63.4** | **53.3** |

Figure 5: **Performance comparison of different data scales for SFT, CFT, Distill, and Mix tuning on Qwen2.5-VL-7B-Instruct**. The top showcases the curves for general benchmarks, and the bottom showcases the curves for temporal benchmarks.

the highest average performance on temporal benchmarks, effectively combining the strengths of all tuning methods. We hypothesize that this is due to the complementary nature of supervision signals: SFT provides direct supervision via ground-truth next events, while CFT and Distill introduce richer semantic feedback through model-generated guidance. This diversity likely enables the model to better generalize in temporal prediction tasks.

**Effect of training data scale.** As illustrated in Figure 5, increasing the NEP training data size beyond 5K samples does not uniformly improve performance across tuning strategies and, in some cases, even leads to slight degradation on both general and temporal benchmarks. We think this saturation is due to several factors. *First*, NEP is specifically designed to enhance temporal reasoning rather than to inject additional general world knowledge. Therefore, we do not expect NEP-only training to consistently improve general-purpose benchmarks as the data scale grows; in this regime, maintaining comparable general performance (i.e., avoiding degradation) is already a desirable outcome. *Second*, to cleanly isolate the effect of the training task itself, we construct NEP data from the intersection of video corpora that have already been used for other training tasks,

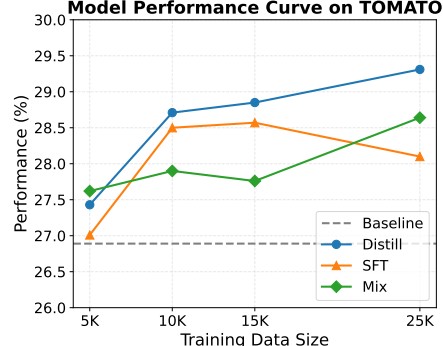

Figure 6: Data-scaling behavior on the TOMATO temporal reasoning benchmark.

Table 3: **Ablation on data construction strategies for NEP tuning**. G-Avg. and T-Avg. represent the average performances of general and temporal benchmarks.

| | General Benchmark | | | | Temporal Benchmark | | | | |
|---|---|---|---|---|---|---|---|---|---|
| | VMME(w/o sub) | MVB | LVB(val) | G-Avg. | TB | TC | SB-R1 | FB | T-Avg. |
| Qwen2.5-VL-7B-Instruct | 59.8 | 65.3 | **55.9** | 60.3 | 35.4 | **73.8** | 37.1 | 52.6 | 49.7 |
| SFT w/o Caption Analysis | **60.9** | 65.6 | 52.9 | 59.8 | 37.1 | 69.4 | 37.6 | 59.5 | 50.9 |
| SFT w. Caption Analysis | 59.2 | **66.5** | 53.4 | 59.7 | **39.9** | 69.9 | **39.1** | **61.3** | **52.6** |

Table 4: Comparison of different training tasks on TOMATO across reasoning types.

| Model | Count | Direction | Rotation | Shape&Trend | Velocity&Freq | Visual Cues | Overall |
|---|---|---|---|---|---|---|---|
| NEP-Distill | 32.53 | 28.54 | 24.48 | 33.63 | 26.67 | 34.29 | 29.31 (**+2.42**) |
| NEP-CFT | 36.99 | 31.76 | 19.58 | 31.39 | 19.52 | 40.00 | 29.04 (**+2.15**) |
| NEP-MIX | 31.51 | 29.53 | 26.92 | 30.49 | 23.33 | 28.57 | 28.64 (**+1.75**) |
| NEP-SFT | 37.50 | 30.52 | 20.80 | 27.58 | 21.42 | 36.43 | 28.57 (**+1.68**) |
| OEQA | 41.10 | 26.05 | 18.53 | 31.39 | 21.43 | 42.86 | 28.50 (**+1.61**) |
| MCQA | 39.73 | 27.30 | 19.93 | 26.46 | 21.90 | 31.43 | 27.63 (**+0.74**) |
| Qwen2.5-VL-7B | 34.93 | 28.29 | 21.68 | 23.32 | 18.10 | 44.29 | 26.89 |
| Captioning | 34.59 | 26.80 | 22.73 | 25.56 | 17.14 | 44.29 | 26.82 (**-0.07**) |

including captioning, open-ended QA (OEQA), and multiple-choice QA (MCQA). Using this core set of videos allows a controlled, task-level comparison under a fixed video corpus, but, in turn, it also means that NEP is trained on a restricted set of video sources. This reduced diversity may limit the room for scaling: when additional NEP samples come from essentially the same narrow distribution, increasing the data size brings smaller gains. *Third*, our experimental protocol deliberately trains NEP in isolation and compares it against models trained solely on MCQA or OEQA, so that we can study "training task vs. training task" under a fixed budget. Because most evaluation benchmarks are formulated in QA format, this design places NEP at a slight disadvantage: NEP-only tuning introduces a shift between the NEP objective and the downstream QA-style usage. For the second and third points, in a practical recipe for pushing state-of-the-art performance, one would collect as much diverse data as possible, combine NEP with QA and captioning data, and carefully schedule when to use which supervision, rather than training with NEP alone. *Finally*, existing temporal benchmarks may under-represent the full benefits of NEP. As discussed in the recent temporal reasoning multimodal evaluation benchmark TOMATO (Shangguan et al., 2024), "existing widely-used temporal reasoning video benchmarks (Liu et al., 2024b; Chen et al., 2024a; Li et al., 2024e) allow models to exploit shortcuts, enabling them to answer correctly using a single, few, or out-of-order frames". In contrast, Figure 6 shows that, on TOMATO, NEP exhibits a more positive data-scaling trend: more NEP training data generally yields better temporal performance, although with some non-monotonic fluctuations.

**Effect of Caption Analysis.** The caption analysis step is designed to improve both the quality and the challenge level of constructed samples. Without this step, we fall back to selecting a sentence boundary from the caption as the text split point. As shown in Table 3, applying SFT to data constructed in a simple way without caption analysis already brings a modest improvement on temporal benchmarks over the untuned baseline (T-Avg.: 49.7 → 50.9), while largely preserving performance on general benchmarks. Furthermore, applying caption analysis on top of the same video corpus yields consistently larger gains in temporal reasoning, increasing T-Avg. from 50.9 to 52.6. These results demonstrate that caption analysis is an effective component of the NEP data construction pipeline.

## 5.3 ASSESSING FINE-GRAINED VISUAL TEMPORAL REASONING CAPABILITIES

Beyond the four temporal benchmarks (TempCompass, TemporalBench, SeedBench-R1, and our FutureBench) in Section 5.1, we further ask whether *next-event prediction* (NEP) improves *genuinely* temporal reasoning rather than exploiting static shortcuts. To this end, we evaluate our models on TOMATO (Shangguan et al., 2024), a recent benchmark explicitly designed to test fine-grained visual temporal reasoning in multimodal foundation models. The benchmark contains 1,484 human-curated multiple-choice questions built on 1,417 videos, including 805 self-recorded or synthesized clips. It covers six temporally focused tasks: *Rotation* (global rotation direction), *Direction* (motion direction), *Velocity & Frequency* (changes in speed or repetition rate), *Shape & Trend* (trajectory shape or overall trend), *Visual Cues* (subtle signals for ordering or timing), and *Action Count* (number of repetitions).

Table 5: **Performance comparison of inductive, deductive, and abductive tasks on temporal benchmarks**. PEP: Previous Event Prediction.

|  | Temporal Benchmark | | | |
|---|---|---|---|---|
|  | TB | TC | SB-R1 | FB |
| Inductive (Video QA) | 36.6 | 74.0 | 35.4 | 58.8 |
| Deductive (NEP) | **38.6** | **74.7** | **39.5** | **61.3** |
| Abductive (PEP) | 38.0 | 66.2 | 31.2 | 55.1 |

Table 6: **Performance comparison of SFT and GRPO with NEP**. G-Avg.: average performance of general benchmarks. Interp.: Interpolation task.

|  | General | FutureBench | | | |
|---|---|---|---|---|---|
|  | G-Avg. | 1-Hop | 2-Hop | 3-Hop | Interp. |
| Instruct | **60.3** | 56.1 | 57.5 | 49.8 | 50.5 |
| NEP w. SFT | 59.7 | 67.6 | 64.2 | 57.7 | 59.3 |
| NEP w. GRPO | 58.2 | **83.8** | **81.3** | **62.7** | **65.2** |

We follow the official TOMATO evaluation pipeline. For each example, the script samples a fixed number of frames, constructs a multiple-choice prompt with the question and answer options, and queries the model once. We then use the official parser to extract the predicted option and compute accuracy, and report scores aggregated by *reasoning type* (the six temporal tasks above). This decomposition allows us to examine whether NEP preferentially improves particular kinds of motion reasoning. For each task, we report the **best** accuracy achieved over training-set sizes from 5K to 25K samples (same as in the previous experiments), as summarized in Table 4. Among all configurations, NEP with the Distill training strategy yields the largest improvement, suggesting it most effectively strengthens temporal reasoning. Consistent with Table 1, training on the multiple-choice QA task leads to lower absolute scores than open-ended video QA on the TOMATO benchmark.

## 5.4 REINFORCEMENT LEARNING WITH NEXT-EVENT PREDICTION

Reinforcement learning (RL) represents an alternative and essential learning paradigm for enhancing reasoning capabilities. To systematically examine the impact of RL-based training of NEP on both general and temporal video understanding, we construct a dedicated training set comprising 2,000 multi-choice QA pairs. This training set is generated using the same pipeline as FutureBench, but is derived from the V1-33K video dataset and restricted to 1-hop and 2-hop extrapolation tasks. Consequently, the 3-hop extrapolation task is treated as an out-of-distribution (OOD) setting, designed to assess model generalization to longer, unseen causal chains. Similarly, the interpolation task (Interp.) presents an additional OOD challenge, requiring the model to reason over fragmented future context. In this experiment, we train the Qwen-2.5-VL-7B-Instruct using Group Relative Policy Optimization (GRPO) (Shao et al., 2024) with the outcome supervision and evaluate its performance across both general and temporally-focused video benchmarks.

**RL generalizes well on FutureBench but degrades performance on general benchmarks.** As shown in Table 6, the GRPO-trained model demonstrates strong performance improvement on in-distribution tasks and generalizes well to OOD tasks, including 3-hop questions and interpolation tasks. These results underscore the effectiveness of RL training in the future event prediction task. However, it is also notable that the RL-trained model suffers from non-trivial performance degradation on general video understanding benchmarks. This suggests that while RL training promotes a reasoning style suited for future event prediction, it may pose inductive biases that hinder generalizability to tasks not requiring future-oriented prediction. Furthermore, we observe instances of reward hacking, wherein RL training with multi-choice QA and outcome supervision may encourage models to exploit superficial patterns, such as lexical similarity between answer options and the question text, to arrive at correct answers. Such behavior deviates from our initial motivation and this shortcut undermines the intended objective of next-event prediction, which is to foster integrated visual perception and causal reasoning. Given these limitations, *we highlight that SFT-style training remains a simple yet efficient approach for training on NEP*.

## 6 CONCLUSION

In this work, we propose next-event prediction, a self-supervised learning task designed specifically to improve temporal reasoning capabilities in MLLMs. By dividing videos into past and future frames, NEP forces models to predict unseen future events, enabling models to implicitly build robust internal representations of causal and narrative dynamics. To study NEP and facilitate research in this area, we created V1-33K, a large dataset of approximately 33,000 video instances that cover a wide range of real-world scenarios and temporal complexities. Furthermore, we proposed FutureBench, a comprehensive benchmark that assesses models' ability to generate logically coherent and causally consistent future event predictions. Experiments show that incorporating NEP significantly improves MLLMs' temporal reasoning capabilities while maintaining their performance on traditional video understanding tasks. We believe that NEP lays a foundation for advancing temporal understanding in MLLMs, bridging the gap between static visual description and temporal event inference.

ACKNOWLEDGMENTS

This material is based upon work supported by the Air Force Office of Scientific Research under award number FA2386-24-1-4011, and this research is partially supported by the Singapore Ministry of Education Academic Research Fund Tier 1 (Award No. T1 251RES2509).

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

## A    APPENDIX: RELATED WORK

### A.1    VIDEO INSTRUCTION-TUNING OF MLLMS

The fusion of vision and language in large models has advanced rapidly from image-focused models like CLIP (Radford et al., 2021) and LLaVA (Liu et al., 2023) to recent video-language models that interpret dynamic visual content leveraging the advanced ability of LLMs (Liang et al., 2024; Tang et al., 2023). Early approaches adapted image-based techniques by fine-tuning LLMs with an extended visual encoder on video frames for observational tasks, such as captioning and question answering; this process is also known as video instruction tuning. For instance, Video-ChatGPT (Maaz et al., 2023) was trained on a large custom 100k video-instruction (Q&A) dataset derived from sources like ActivityNet-200 to enable detailed video conversation and reasoning. Video-LLaVA (Lin et al., 2023) used mixed image/video datasets focusing on unified visual QA across modalities. Video-LLaMA (Zhang et al., 2023) utilized large-scale video/image caption datasets (e.g., Webvid-2.5M, LLaVA-CC3M) for pre-training and various instruction datasets (MiniGPT-4, LLaVA, VideoChat) for fine-tuning, targeting video-to-text generation and audio-visual instruction following. LLaVA-OneVision (Li et al., 2024a) leveraged large-scale image instruction data (covering QA, OCR, Math, etc.) aiming for cross-scenario transfer to video understanding tasks like referring analysis, and LLaVA-NeXT-Interleave (Li et al., 2024c) compiled the diverse M4-Instruct dataset (1.18M samples across 41 datasets) for interleaved multi-modal QA and captioning across images, video frames, and 3D views. Even models built by companies, like Qwen2.5-VL (Bai et al., 2025), Kimi-VL (Team et al., 2025) and Aria (Li et al., 2024b) primarily build upon datasets focused on description, QA, or dialogue generation based on observed content.

However, their training data and objectives are predominantly observational, describing or explaining visible content, rather than predictive. Our work differs by introducing a predictive objective, next event prediction, to explicitly train the model's temporal reasoning abilities. This aligns with the goal of modeling world dynamics, extending beyond static understanding of frames to reasoning about how scenes evolve over time. While prior instruction-tuned MLLMs were not primarily designed for complex temporal reasoning, their advanced video perception abilities provide a strong foundation, paving the way for our work to further explore and advance temporal reasoning mechanisms in MLLMs.

### A.2    FUTURE PREDICTION IN COMPUTER VISION

Anticipating the future has been studied in computer vision in various forms. A significant body of work focuses on action prediction tasks, which frame future prediction as a classification problem. The goal is to improve video encoder by predicting the next discrete action label, typically from a predefined set of (verb, noun) pairs, based on observed video frames (Miech et al., 2019). Within this paradigm, various methodologies have been explored to improve predictive accuracy. For instance, Liu et al. (2024a) define the task into action recognition and introduce a Next Action Prediction (NAP) objective, analogous to next-token prediction in language models, to explicitly model the statistical correlations between consecutive actions. Other approaches focus on incorporating richer input modalities. Beedu et al. (2024) propose the M-CAT model, which demonstrates that using text-based descriptions of objects and actions as an additional input modality can provide crucial context, thereby enhancing the model's ability to classify the correct future action. Methodological improvements have also been achieved through advanced training strategies. For example, Tran et al. (2021) utilize a knowledge distillation framework where a student model, which only observes past frames, is trained to predict the future-aware representations of a teacher model that has access to both past and future frames.

Another major paradigm uses future prediction as a self-supervised objective for representation learning. Here, the goal is not the prediction itself, but to pre-train a powerful, general-purpose video encoder for various downstream applications. For example, Liu et al. (2025) propose a framework that reconstructs masked patches of the current frame in a latent space. To guide this reconstruction, it employs a "sandwich sampling" strategy that leverages temporal correspondence from both past and future frames, explicitly avoiding the low-level detail required by pixel-space reconstruction (Liu et al., 2025). Similarly, Assran et al. (2025) introduce V-JEPA 2, a joint-embedding predictive architecture that learns a world model by predicting the representations of masked video segments in

a learned feature space. While V-JEPA 2 achieves state-of-the-art results on the downstream task of action anticipation, its primary contribution is the learned powerful video encoder.

Distinct from both classification and representation learning, Hosseinzadeh & Wang (2021) propose a method for video captioning of future frames where the goal is to generate a sentence describing a likely future event. They propose a disjoint approach: in first stage, it first predicts the video representation of unobserved frames; and in the second stage, it feeds these predicted features into a dedicated captioning translator to convert representation into the final textual description.

These works typically define the "future" at short horizons, e.g., the next frame or next action, and optimize low-level objectives, e.g., representation learning, which has proven effective for video encoder pretraining. Our work is distinct in targeting high-level, semantic future-event prediction expressed in natural language. This NEP task requires MLLMs to perceive temporally grounded visual evidence from the vision module, integrate it with pretrained commonsense knowledge in the LLM module, and generate open-ended outcomes in natural language. Because the goal differs, so does the focus of optimization: prior methods mainly update the visual encoder, we freeze the encoder and fine-tune the vision–language projector and the LLM, enhancing multimodal temporal reasoning.

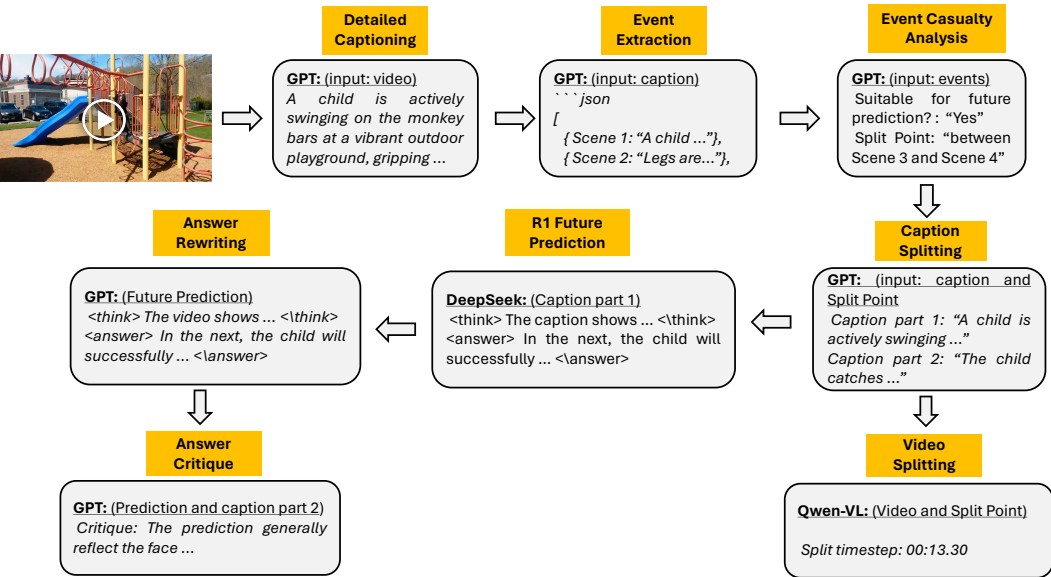

Figure 7: Data Construction Pipeline.

# B  DETAILED DATA CONSTRUCTION PIPELINE

## B.1  V1-33K CONSTRUCTION

**Fact Translation.**  In this initial stage, visual content is translated into a textual format to serve as the foundation for further processing. For every video, we use a Vision-Language Model (VLM) to generate a detailed caption that comprehensively describes the visual facts. This conversion from visual to textual data ensures that the strong text-based reasoning capabilities of open-source large language models (LLMs) can be leveraged.

**Analysis.**  Given the fact that current models exhibits stronger reasoning capabilities when working with text, we feed the detailed captions into a LLM. The LLM performs two critical tasks:

- *Scene Identification:* It dissects the caption to extract and delineate distinct scenes.
- *Causal Analysis:* It evaluates the causal relationships between scenes and identifies an optimal split point where the context from preceding events is strong enough to predict what comes next.

This step establishes a structured understanding of the video, which is crucial for effective segmentation. The prompt used in this step can be found in Appendix E.2.

**Segmentation.** Using the optimal split point determined during the Analysis stage, we partition both the original video and its caption into two parts. The first part of the video, which contains the initial events, serves as a clear input for the video reasoning model, ensuring that the video reasoning is based on established facts. The second segment is reserved as the ground truth for evaluating the model's predictions. The prompt used in this step can be found in Appendix E.3.

**Reasoning & Critique.** One promising approach to rapidly enhance video reasoning is through Long CoT supervised fine-tuning. In our dataset, we leverage the output of a text reasoning model to facilitate this process. Specifically, the text reasoning model (DeepSeek-R1) processes the first part of the caption, recording its reasoning process and generating predictions for future events. Recognizing that textual reasoning can sometimes introduce errors, we subsequently employ an additional LLM to critically evaluate both the reasoning process and the resulting predictions. This approach draws inspiration from recent advances in critique fine-tuning (CFT), where models learn to critique noisy responses, pecifically the reasoning and predictions, rather than simply imitating them through SFT. By doing so, we ensure that only robust reasoning informs the final training of the MLLM, ultimately boosting its overall performance. The prompts used in this step can be found in Appendix E.4 and E.5.

The data processing pipeline is outlined below. We employ DeepSeek-R1 Guo et al. (2025) for the Future Prediction step and Qwen2.5-VL-72B-Instruct for Video Splitting, while using the O3-mini Achiam et al. (2023) for all other steps. The prompts used at each stage are critical for high-quality data processing. We have made efforts in manually testing a wide range of hand-written prompts and playing with the API.

Table 7: Statistics and distribution of data source for Extrapolation and Interpolation in FutureBench. #Total indicates the total size of each subset.

| Data Source | Extrapolation | | | Interpolation |
| --- | --- | --- | --- | --- |
| | 1-Hop | 2-Hop | 3-Hop | |
| **#Total** | 173 | 193 | 201 | 489 |
| YouTube | 48.0% | 37.3% | 45.3% | 51.9% |
| ActivityNet | 23.1% | 31.6% | 24.9% | 23.5% |
| YouCook2 | 11.6% | 10.4% | 10.0% | 8.2% |
| NextQA | 8.7% | 10.4% | 10.0% | 8.2% |
| Charades | 8.6% | 10.3% | 9.8% | 8.2% |

### B.2 FUTUREBENCH DETAILS

We discuss the details of FutureBench construction in Section 4.2. Note that the videos used in FutureBench have no overlap with V1-33K to ensure fair evaluation despite the same curation pipeline. FutureBench also involves videos from diverse sources. The final statistics of FutureBench and distribution of the data source are shown in Table 7.

### B.3 HUMAN-IN-THE-LOOP QUALITY REVIEW

Following automatic generation, all QA items undergo a verification and filtering process. Samples deemed too trivial – **such as those with answers directly inferable from a single frame, or with implausible distractors using simple commonsense** – are discarded. QA pairs requiring minor corrections are edited to ensure semantic coherence and alignment with the underlying video narrative. This human-in-the-loop review process allows us to maintain high annotation quality while leveraging GPT-4 to scale data generation efficiently Miao et al. (2024).

As a result, FutureBench comprises a total 1056 carefully curated QA pairs spanning both extrapolation and interpolation subtasks. To assess the benchmark's quality and highlight both visual perception and temporal reasoning, we evaluate a strong reasoning model, o4-mini, on the text-only version of questions, excluding any visual input. The model achieves an accuracy of 32.0%, suggestion that **even advanced reasoning capabilities alone are insufficient for consistently solving the**

**tasks**. This finding reinforces the critical role of visual perception in solving future event prediction in FutureBench

## C  QUALITATIVE EVALUATION OF DATA AND BENCHMARK

### C.1  QUALITATIVE STUDY OF FUTUREBENCH

In here we provide the qualitative studies of the FutureBench.

For the sample shown in Figure 8,option A is correct because after the observed scenes (where the digits and straight lines are drawn), the next logical step in the timeline is to modify the drawn '2' s by adding a 'C' shape on top of each one. This action initiates the transformation of the numbers into elements that will later become detailed bird-like figures, fitting neatly between the earlier steps and the final completed image.

For the sample shown in Figure 9, option A correctly predicts the intermediate events: a chaotic moment where the man tumbles on the countertop, followed by the woman's concerned reaction and her cracking an egg into a bowl, which bridges logically between the observed scenes and the final scene of presenting the cake. Option B distracts by describing a calm, uneventful sequence that does not match the dynamic and slightly chaotic tone of the video. Option C introduces conflict that is not supported by the context of collaborative, playful baking, and Option D describes unrelated actions (ignoring the kitchen and chopping vegetables) that do not bridge to the final cake presentation.

For the sample shown in Figure 10, option D correctly bridges the gap by introducing the sauteing of fresh ingredients, which establishes the cooking of the dish, followed by the careful plating that logically leads into the final garnished presentation. Option A incorrectly emphasizes rolling and cutting dough again, repeating events already observed and omitting the important cooking phase. Option B ignores the gradual progression of cooking and assembling the dish by suggesting an immediate processing of the dough. Option C, although mentioning dough cutting and feeding, lacks the essential steps of sauteing and final plating that would lead to the garnished final scene.

**Question:** Based on the given video, predict future events and fill in the intermediary action that should occur next prior to the final scene, which shows the completed bird-like figures with detailed wings. Identify the missing step: 1. [?] 2. The video concludes with a text overlay inviting viewers to try drawing the finished birds. What should the missing intermediary event be?

**Seen videos**

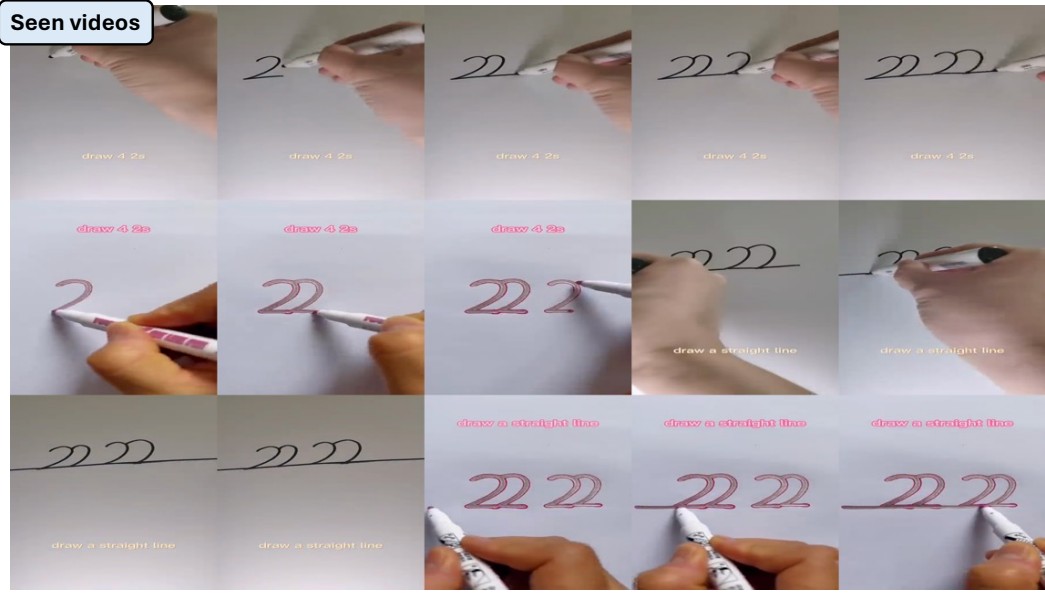

A: A hand holding a black marker draws a 'C' shape on top of each '2' to begin transforming the numbers into bird-like figures.

B: A hand holding a pink marker draws two 'W' shapes beneath the figures to indicate the start of wing formation.

C: A hand holding a black marker draws two curved lines on each '2', forming bird-like shapes with implied bodies.

D: A hand holding a pink marker adds bean-like eyes and beaks to the figures, finalizing the bird faces.

**Unseen videos**

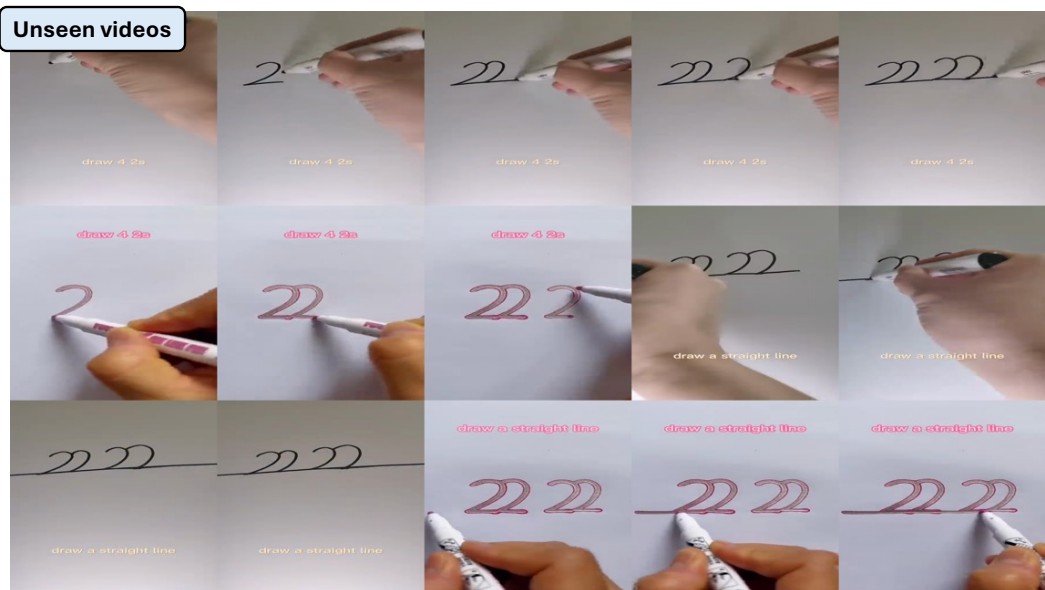

Figure 8: The visualization of FutureBench ID: `lq9v0XXFg6E`.

**Question:** Based on the given video, predict future events and fill in the potential events in the following future event slots, assuming the video ends as shown later: 1. [?] 2. [?] 3. [The man presents a tall, multi-layered cake with a sparkling candle].

**Seen videos**

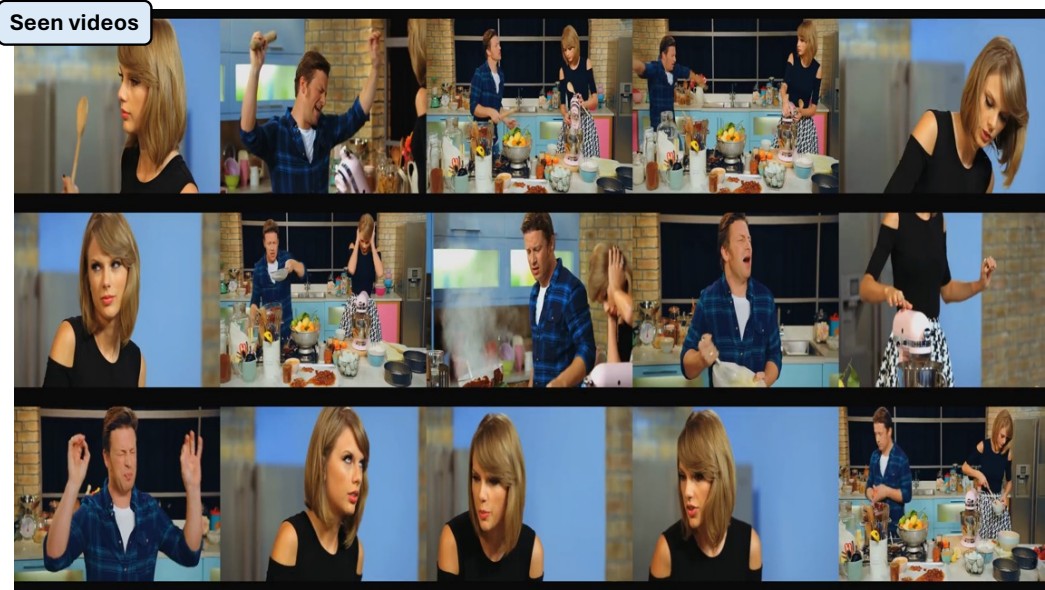

A: 1. **The man performs a flip and tumbles over the countertop with flailing legs; 2. The woman appears concerned and cracks an egg into a bowl.**

B: 1. The woman adjusts her hair while the man sifts flour without incident; 2. Both share a quiet, calm moment sorting ingredients.

C: 1. The man accidentally spills a bowl of chocolate leading to silence; 2. The woman scolds him and leaves the kitchen.

D: 1. The woman glances at her phone ignoring the kitchen; 2. The man starts chopping vegetables intensely.

**Unseen videos**

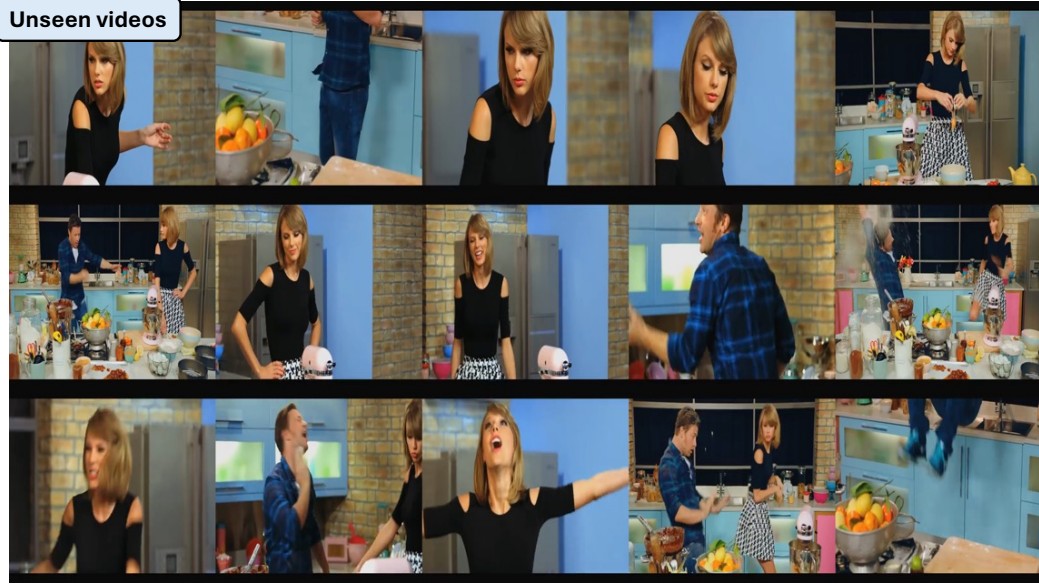

Figure 9: The visualization of FutureBench ID: v_HOTCR1uIaBM

**Question:** Based on the given video, predict future events and fill in the potential events in the gap before the final scene (a plated pasta dish is garnished and placed on the counter for serving). Complete the future sequence by selecting the pair of intermediate events that best bridges the observed scenes (1 to 9) to the final scene: 1. [?] 2. [?] 3. In the final scene, two white bowls of pasta are garnished by chefs with a blue finishing tool and a drizzle of green sauce, and the plated pasta is placed on a counter ready for serving.

**Seen videos**

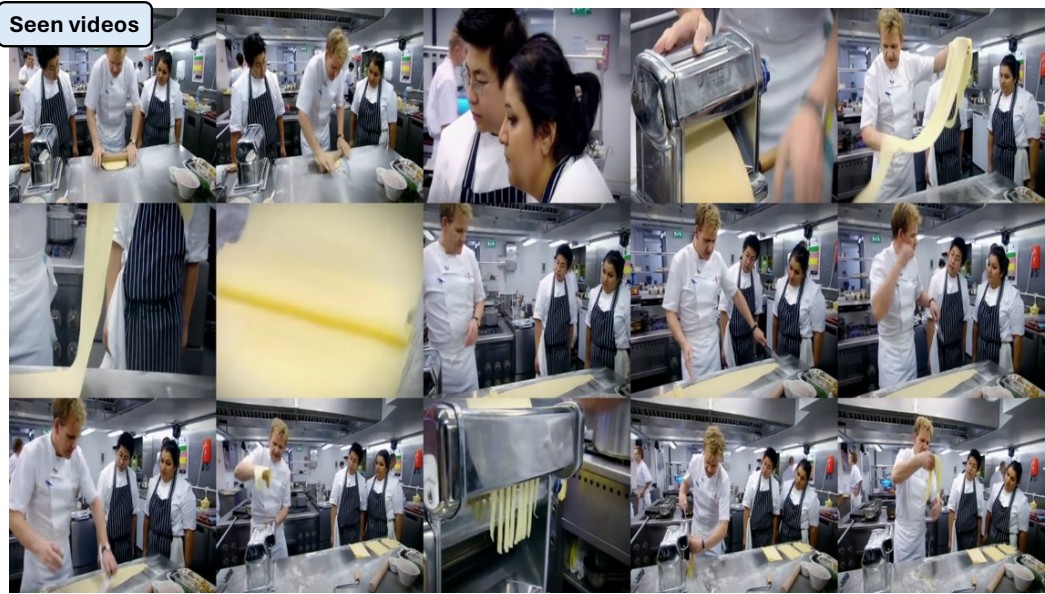

A: 1. A chef rapidly rolls out dough and cuts it into shapes using a knife; 2. Chefs exchange ingredients across the kitchen countertop before switching to dough feeding.

B: 1. Chefs feed the dough directly into a pasta machine without further preparation; 2. The pasta is immediately tossed with herbs in a bowl.

C: 1. A chef demonstrates the dough-cutting technique by first slicing the dough into precise shapes; 2. The cut dough is run through the pasta machine and its output is briefly inspected by other chefs.

**D: 1. A chef sautés fresh ingredients including green herbs and vegetables in a metal pan on the stove while explaining the cooking process; 2. A chef uses a metal pan to transfer the mixture into a white bowl and then applies final plating techniques, setting the stage for the dish to be garnished in the final scene.**

**Unseen videos**

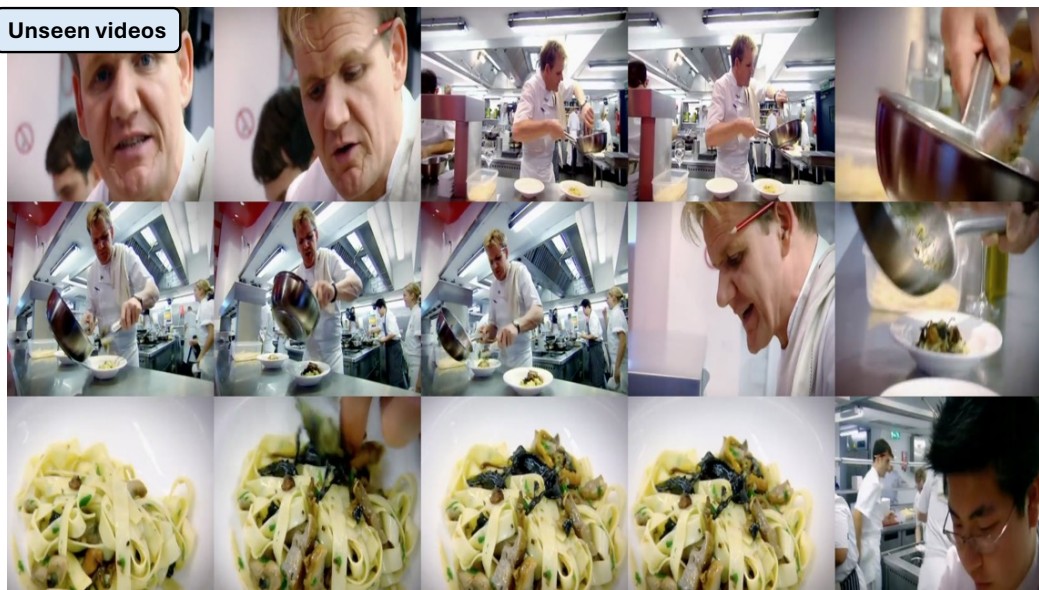

Figure 10: The visualization of FutureBench ID: v_3VzXH3o88mw

## C.2 QUALITATIVE ANALYSIS OF FUTURE EVENT PREDICTION TRAINING DATA

We qualitatively examine Next-Event Prediction (NEP) training pairs to verify that the future-event captions are (i) linguistically coherent and (ii) tightly grounded in the observed past video, rather than merely reflecting generic language priors. As shown in Figure 11, the past video segment shows people launching a canoe and paddling along a calm river surrounded by dense trees and cliffs. The corresponding future-event text continues this storyline: the canoe keeps moving downstream and the people begin fishing on the river. Predicting such a future requires the model to track the ongoing activity over multiple frames and infer a plausible next step in the same physical environment (continuing to paddle, then fishing), instead of describing the already-seen content or resorting to a template such as "after boating, people usually . . . ".

Another example is shown in Figure 12. In video part, the child approaches an arcade bowling machine, inspects the interface, and waits while the lane and pins are being prepared. In text part, the future-event caption describes the child retrieving a red ball, rolling it down the lane, and watching the pins being knocked over. Here, the future description is a concrete consequence of the earlier interaction: the child's initial exploration of the machine, the resetting of pins, and their focused posture collectively imply that the next stage is actively playing the game. This again reflects the core NEP objective-bridging past and future segments via causal and temporal reasoning about how human actions unfold over time.

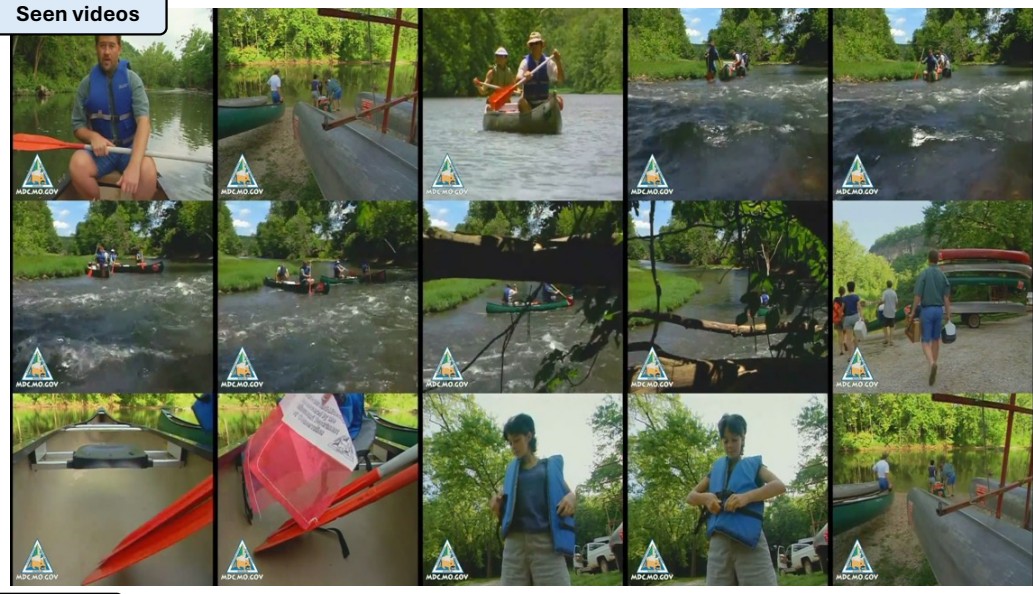

**Seen videos**

**Future**

As the trip continues, two people in blue life jackets paddle a green canoe farther downstream. They move past steep, tree-covered banks, keeping a steady rhythm as the calm river reflects the surrounding greenery. The canoe drifts in and out of light current, but the pair remain in control and continue their relaxed journey.\n\nFarther along, one person in the group pauses to fish from the canoe near a rocky stretch of riverbank, casting a line toward the quieter water while others keep paddling nearby. The canoes take turns gliding through gentle riffles and returning to calmer pools, with small waves forming around the bows.\n\nMore canoes join together into a loose group, with paddlers in different colored life jackets chatting and paddling at an easy pace. The river widens slightly, the sky clears, and sunlight glints off the moving water. The final shots follow the canoes from a distance as they continue downstream, cutting through the ripples and disappearing around a bend in the peaceful, forested river.

Figure 11: The visualization of training data.

**Seen videos**

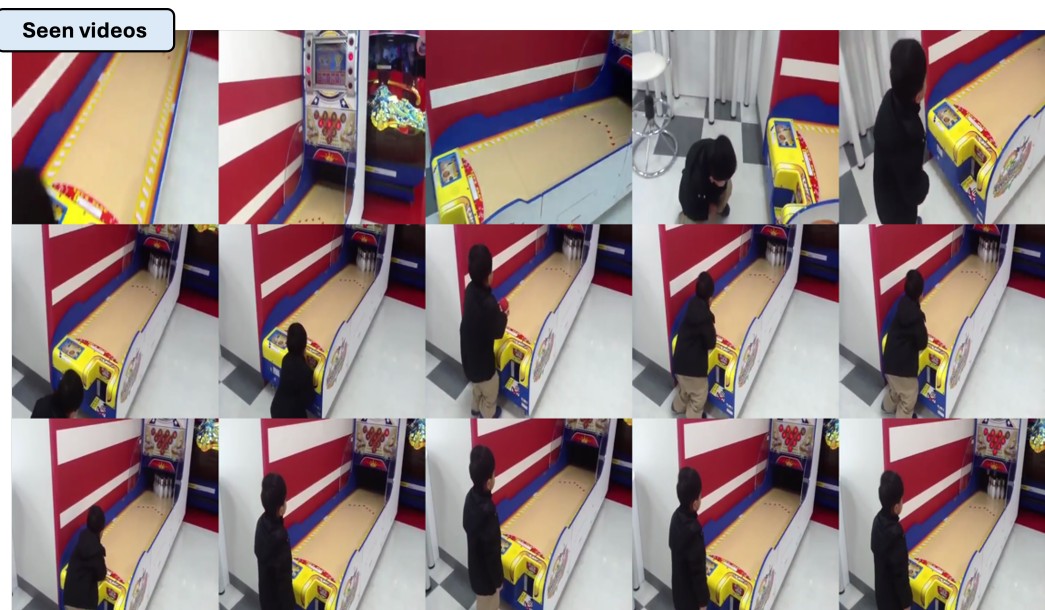

**Future**

The child continues to focus on the arcade bowling game, picking up a red ball and rolling it down the lane toward the freshly reset pins. After each roll, the ball travels along the smooth lane, taps or knocks over some of the pins, and the machine briefly lights up as it clears and resets them for the next turn. The child watches each roll closely, stepping back to see the result and then moving forward again to retrieve another ball. Over several turns, the child repeats this pattern of rolling, observing the pins, and preparing for the next attempt, remaining absorbed and determined throughout the game.

Figure 12: The visualization of training data.

## C.3 Qualitative Analysis of Model Prediction Results

For the future prediction example in Figure 13, the predictions from Qwen2.5-VL-7B and its NEP-trained variant are shown below.

The NEP model's analysis is correct because it aligns with both the video and the task definition: it understands that the model should predict a plausible future event based on the ongoing, methodical laundry behavior, and it interprets option A as a natural continuation of that pattern (carefully handling and sorting clothes, pausing thoughtfully). It also correctly treats "taking the mug and leaving" as part of the observed sequence, not something to be re-described as a separate future event. In contrast, the Qwen2.5-VL-7B prediction is incorrect because it downplays the sorting behavior, overemphasizes the mug, and effectively treats "grab mug and leave" as the key future event, which misinterprets what the options are intended to capture. As a result, it rejects A on overly narrow grounds and incorrectly chooses C, even though C conflicts with the broader behavior shown in the description and with the intended notion in the question.

**Question:** Based on the given video, predict the future events and fill in the potential event in the following gap: 1. [?] 2. The person retrieves a mug from the top of the washing machine, walks towards the door, opens it, steps out of the laundry room, and closes the door behind them. Which of the following descriptions best fills the gap for the missing future event?

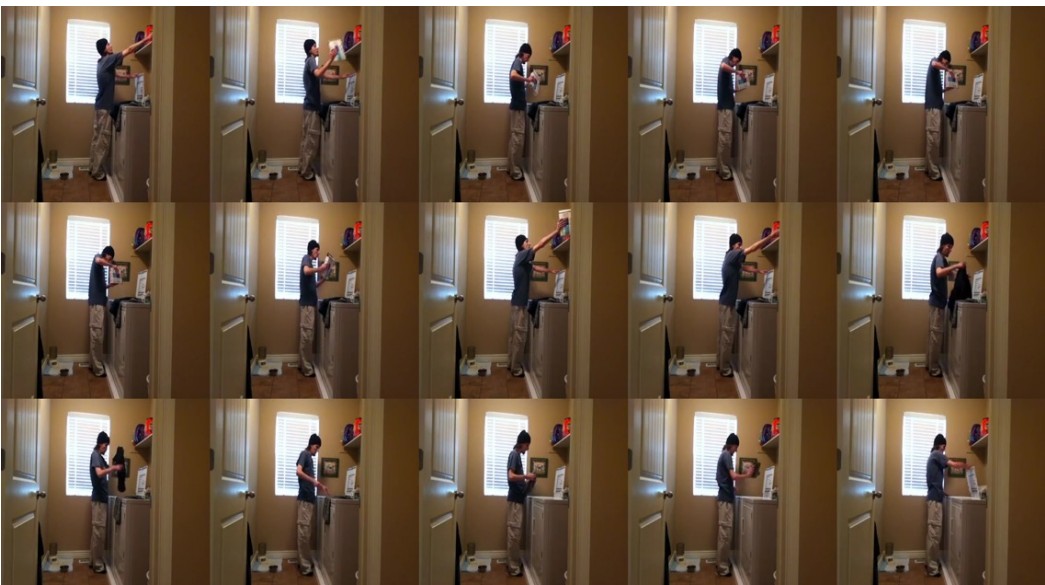

A: The person methodically sorts through the clothes on the dryer – neatly folding some items and setting others aside – and then pauses to touch their head in a moment of thoughtful decision.

B: The person pauses to touch their head immediately, then hurriedly shuffles the clothes from the dryer into a basket without folding.

C: The person directly retrieves the mug from the washing machine without addressing the pile of clothes, then walks toward the door in a rush.

D: The person returns to the detergent shelf to fetch another container and, while doing so, intermittently glances at the clothes on the dryer without sorting them.

Figure 13: The visualization of FutureBench ID: 511AM

**NEP Trained Model Output for Task in Figure 13**

Let's break down the video description to predict what happens next. The person is in a laundry room, organizing items on shelves above the washing machine. They handle a blue container, a white box with a red label, and a black garment. After placing these back, they pick up a dark piece of clothing from the dryer and examine it before putting it back. Then they take a mug from the washing machine, walk out, and close the door.\n\nFirst, considering the sequence of actions: organizing items, handling clothes, retrieving the mug. The person seems methodical but maybe slightly distracted, as they pause to touch their head. The presence of the detergent shelf suggests they might need more supplies, but after taking the mug, they leave. Since they closed the door upon exiting, perhaps they're done with the laundry task. However, the pile of clothes on the dryer hasn't been addressed yet. \n\nLooking at option A: The person is sorting clothes, pausing thoughtfully. This aligns with the initial actions-organizing and handling items. But after that, they move to the mug. So maybe after dealing with the mug, they might return to the clothes. But since they left the room, perhaps the next step is taking the mug elsewhere, like the kitchen. Alternatively, if they left the laundry room, maybe they're done, but the clothes are still there. However, the prediction should follow the immediate next steps. Since they took the mug and left, the immediate future would be them using the mug elsewhere, possibly in another room. But the question asks about future events in the video. If the video ends with them leaving, maybe the next scene isn't shown, but based on the described actions, the person has completed their tasks in the laundry room. Therefore, the prediction should focus on their departure and what they might do next with the mug. However, the original caption says they closed the door, so maybe the video ends there. But the user wants a prediction beyond the given caption. Wait, the caption describes the entire video up to the point where they exit. So the prediction is about what happens after they exit. But the user might want a prediction based on the last action: retrieving the mug and leaving. So possible outcomes: they might use the mug for something else, like drinking coffee, or place it somewhere else. Alternatively, maybe they forgot something and come back, but that's less likely. Given the methodical nature, perhaps they're done and moving on to another task. Since the video ends with them closing the door, the prediction could be that they proceed to another area, like the kitchen, to use the mug. Alternatively, if the mug was part of a routine, maybe they put it away. But without more context, the safest bet is that they exit the laundry room, having completed their tasks, and the next action is unrelated to the laundry.

**NEP Trained Model Output for Task in Figure 13 (Cont.)**

So the answer should reflect that they've finished and are moving to another activity, possibly involving the mug. But since the caption ends with them leaving, the prediction is about what they do next. Maybe they take the mug to another room, like the kitchen, to have a drink. Alternatively, they might have forgotten something and return, but that's speculative. The most logical conclusion is that they exit, having completed their laundry-related tasks, and the mug is now being used for its intended purpose elsewhere. So the answer should mention them proceeding to another area, perhaps preparing a drink, using the mug, and maybe the scene transitions to show them in another room. But since the prediction needs to be concise, focusing on the immediate next step after exiting. So the answer is that they will likely carry the mug to another room, such as the kitchen, to use it, possibly indicating the end of their laundry chore and moving on to another task. Alternatively, they might place the mug back in the laundry room, but since they closed the door, that's less likely. Therefore, the prediction is they exit, having completed their tasks, and the mug is now being used elsewhere. The answer should be confident but allow for uncertainty if needed. Since the person paused to touch their head, maybe they were thinking about something else, but the main action is leaving with the mug. So the prediction is that they proceed to another area, likely to use the mug, completing their laundry-related activities.\nThe answer is A.

**Qwen2.5-VL-7B Output for Task in Figure 13**

Let's analyze the sequence of actions in the video and the options provided:\n\n1. The person is initially seen reaching up to the shelf above the washing machine.\n2. They retrieve several items (such as detergent and a container), briefly inspect them, and place them back on the shelf.\n3. They then move to the dryer, pick up a dark piece of clothing, look at it for a moment, and put it back without folding or sorting the rest of the pile.\n4. Instead of continuing with the laundry, they shift their attention to the washing machine, retrieve a mug from inside, and immediately head toward the door.\n5. They walk out of the laundry room with the mug and close the door behind them, leaving the pile of clothes on the dryer unaddressed.\n\nNow, let's evaluate each option:\n\nA. This option describes the person methodically sorting through the clothes on the dryer and pausing thoughtfully. While the person does briefly handle a piece of clothing, there is no clear indication that they engage in a sustained or systematic sorting process, so this does not fully match the observed sequence.\nB. This option suggests that the person pauses to touch their head immediately and then hurriedly shuffles the clothes into a basket without folding them. This rushed, bulk-moving behavior is not supported by what we see in the video.\nC. This option states that the person retrieves the mug from the washing machine without properly dealing with the pile of clothes and walks toward the door to leave, which closely matches the observed behavior: they only briefly interact with the clothes and then prioritize taking the mug and exiting the room.\nD. This option mentions that the person returns to the detergent shelf to fetch another container and keeps glancing at the clothes without really acting on them, which does not align with the fact that they leave the room right after taking the mug.\n\nGiven this analysis, the most accurate description of the future events is option C.\n\nThe answer is C.

Another example is shown in Figure 14. In the dog-grooming video, the base model (without NEP) prefers option A, which inserts yet another close-up grooming scene. This is textually plausible given the prompt ("dog grooming & pet supplies") and reflects a strong prior that such videos should keep showing grooming actions, but it ignores the **temporal role** of the missing clips: they are supposed to be the last interior segments immediately before the closing exterior storefront shot, after multiple grooming interactions have already been shown. In contrast, the NEP-trained model selects option B, which focuses on a specific pet bed with brochures followed by a wide product-shelf shot. This sequence is less "obvious" from language priors alone, but it matches the actual chronology and structure of the advertisement: the video first establishes the service (grooming), then transitions to product displays, and only then returns to the exterior sign for branding. The NEP model thus demonstrates sensitivity to temporal structure and previously observed content (it avoids re-describing past grooming events and chooses scenes that function as a bridge to the final shot), rather than simply following generic textual expectations about what typically happens in a dog-grooming video.

**Question:** Based on the given video, predict future events and fill in the potential events in the following sequence: 1. [?] 2. [?] 3. [A final exterior shot of the building with the 'DOG GROOMING' sign on a teal background]. Which option correctly represents the intermediate scenes that would logically occur between the observed scenes and the final scene?

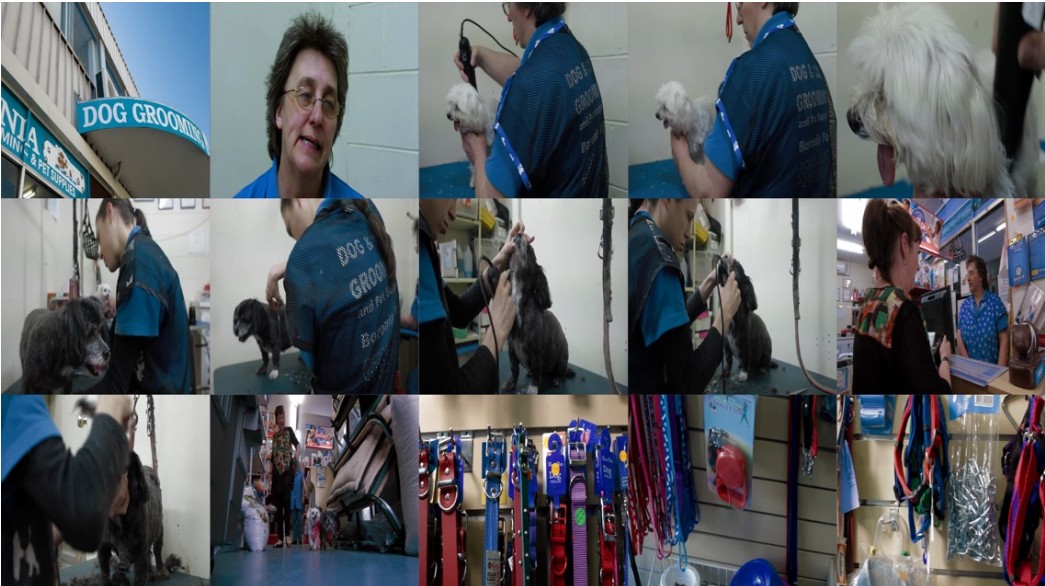

A: A detailed close-up of colorful dog collars displayed on a rack with price tags; then a transition to a person in a blue shirt demonstrating grooming on a small white dog using electric clippers in an indoor setting.

**B: An expansive product display featuring a pet bed with embedded brochures promoting Flea-Free Dog Bed and MultiMat; followed by a comprehensive shelf scene showcasing an array of pet supplies including dog toys, grooming tools, and food packages.**

C: A counter conversation between a person with dark hair in a ponytail and another staff member inside a store; then a quick montage of various grooming tools laid out on a table.

D: An outdoor shot panning across different parts of the building's exterior; then a brief cut to a montage of pet supplies being stocked on shelves.

Figure 14: The visualization of FutureBench ID: `v1-3-train_val-v_J6T9pokWX1E`

**NEP Trained Model Output for Task in Figure 14**

Let's start by breaking down the video provided. The video begins
with the exterior of Boronia Dog Grooming & Pet Supplies,
establishing the location. Then it moves indoors to show staff
members grooming dogs, highlighting their uniforms with the company
name. There are scenes of different dogs being groomed, interactions
between staff, and displays of pet products like collars, leashes,
and beds. The final scene mentioned is an exterior shot of the
building with the 'DOG GROOMING' sign.\n\nTo predict future events
after the final scene described, I need to consider the flow of the
video. The video seems to be promotional, showcasing the services and
products of the grooming salon. Typically, such videos might end with
contact information, a call to action, or a return to the exterior to
reinforce the business details. Since the last scene is an exterior
shot, the next logical step could be displaying the business hours,
phone number, or website. Alternatively, they might show happy pets
and satisfied customers leaving, but since the final scene is already
exterior, maybe adding text overlays with key info.\n\nLooking at the
existing structure: exterior to interior staff grooming to product
displays to exterior again. The video might loop back to the exterior
with additional info. The initial exterior had a sign with 'DOG
GROOMING', so perhaps adding the full address, phone number, or
opening hours. Alternatively, a closing shot with the logo and
contact details superimposed.\n\nBut the user wants future
predictions based on the current scenes, not necessarily the very
end. However, the final scene described is the exterior shot. So
maybe the prediction should focus on what happens immediately before
that final shot. Wait, the caption says the video concludes with that
exterior shot. So the future prediction would be after that? But the
task is to predict future events based on the scene, so maybe the
user expects the prediction to be about what comes after the
described scenes, but according to the caption, the video ends with
the exterior shot. Hmm, this is a bit confusing.\n\nWait, the user's
instruction says: \"Based on the scene, analyze and predict future
events. Provide concise, natural, and confident prediction about the
future events of the video.\" The caption describes the entire video
up to the conclusion. So perhaps the user wants the assistant to
imagine that the video continues beyond the described scenes. But
according to the caption, the video as described ends with the
exterior shot. Therefore, the prediction should be about what would
happen next if the video continued beyond that
point.\n\nAlternatively, maybe the user made a mistake in the
caption, thinking that the video continues, but according to the
description, it's concluded. This is a bit ambiguous.

**NEP Trained Model Output for Task in Figure 14 (Cont.)**

However, given the task, we have to work with the information provided.\n\nAssuming the video is partway through and the final scene is yet to come, but according to the caption, the video has already concluded with that exterior shot. Therefore, perhaps the user intended the final scene to be part of the video, and the prediction is about what would follow naturally after the scenes described.  For example, after showing the exterior, maybe transitioning to contact details or a logo.\n\nBut the user's instruction says to predict future events based on the scene. If the video as described ends with the exterior shot, then there are no more scenes to predict. However, maybe the user expects the assistant to infer that after showcasing the grooming and products, the next logical step is to display contact information or a call to action. Alternatively, the video might show a satisfied customer or a dog leaving the salon, but since the final scene is exterior, perhaps adding text with the business details.\n\nAlternatively, considering the structure of promotional videos, after showing the products and services, they often end with contact details. So the prediction could be that the video will display the business's contact information (phone number, website) over the exterior shot, or transition to a logo with the tagline.\n\nBut since the user wants the prediction to be based on the scenes described, which include the exterior, interior grooming, product displays, and staff interactions, the next logical step after these scenes would be to wrap up with contact info. However, the final scene described is the exterior shot. Therefore, the prediction should be about what would come after that, but according to the caption, the video ends there. Therefore, perhaps the assistant should infer that the video would conclude with contact details superimposed on the exterior shot.\n\nAlternatively, maybe the video is structured to show the exterior, then move into the interior, then products, then staff, then back to exterior with contact info. But the caption says the video concludes with the exterior shot. So perhaps the prediction is that after the exterior shot, the video might display contact information for Boronia Dog Grooming, such as a phone number or website, possibly with a voiceover or text overlay.\n\nBut the user wants the prediction to be concise, natural, and confident, avoiding references to reading captions. So phrasing it as observing the scene, like: \"The video will likely conclude by displaying contact details for Boronia Dog Grooming, such as a phone number or website, overlaid on the exterior shot to encourage viewers to visit.\"\n\nHowever, the user's example response uses options from a multiple-choice question. The answer was \"The answer is B\".\n

**Qwen2.5-VL-7B Output for Task in Figure 14**

```
To determine the correct sequence of events that logically follows
the given video clips, let's analyze the content of each clip and the
options provided:\n\n1. The first clip shows the exterior of
\"BORONIA DOG GROOMING & PET SUPPLIES\" with a focus on the 'DOG
GROOMING' sign.\n2. The second clip transitions to a woman in a blue
shirt grooming a small white dog with electric clippers.\n3. The
third clip shifts to a woman in a black shirt interacting with a
customer at a counter.\n4. The fourth clip shows a woman grooming a
black dog with clippers.\n5. The fifth clip displays a variety of dog
collars and leashes on a rack.\n6. The sixth clip showcases a pet bed
with brochures for 'Flea-Free Dog Bed' and 'Multimat'.\n7. The
seventh clip presents a shelf with various pet supplies, including
dog toys, grooming tools, and food packages.\n8. The eighth clip
shows a woman grooming a small white dog with clippers.\n9. The ninth
clip features a woman grooming a black dog with clippers.\n10. The
tenth clip displays a variety of dog collars and leashes on a
rack.\n11. The eleventh clip shows a counter conversation between two
people.\n12. The twelfth clip presents a quick montage of grooming
tools on a table.\n13. The final clip shows a final exterior shot of
the building with the 'DOG GROOMING' sign on a teal
background.\n\nNow, let's evaluate the options:\n\nA. This option
starts with a close-up of dog collars, which fits the context of the
video as it shows pet supplies. It then transitions to a grooming
session, which also aligns with the video content. This option seems
logical and follows the sequence well.\n\nB. This option starts with
a product display of a pet bed, which is not directly related to the
initial exterior shot of the grooming salon. It then moves to a shelf
scene, which is also not directly connected to the initial grooming
session. This option does not follow the sequence logically.\n\nC.
This option starts with a counter conversation, which is not directly
related to the initial exterior shot of the grooming salon. It then
moves to a montage of grooming tools, which is more aligned with the
video content but does not follow the sequence logically from the
initial exterior shot.\n\nD. This option starts with an outdoor shot
of the building, which is similar to the initial exterior shot but
does not provide any direct connection to the grooming session. It
then moves to a montage of pet supplies, which is not directly
related to the initial exterior shot.\n\nGiven the analysis, the most
logical sequence that follows the initial exterior shot and leads to
the final exterior shot is Option A.\n\nThe answer is A.
```

In the following, we discuss the failure pattern of the NEP-trained model. As shown in Figure 15 and Figure 16.

As the model response for example in Figure 15, the model overlooks the two explicit future anchors specified in the textual question and fails to precisely localize the step to the stage where ingredients are still being incrementally added. Instead, it relies on generic cocktail-making knowledge and treats "shaking and plating the bottle" as a plausible subsequent action, thus incorrectly choosing option B.

Similarly, Similarly, in the hair-styling example (shown in Figure 16), the model again produces a reasonable future continuation at the narrative level—it imagines the person continuing to section the hair, re-apply oil, and use the spray bottle for final touches—but fails to align this prediction with the way the question specifies future events. Instead of leveraging the discriminative visual anchors that actually appear later in the video (the yellow hair clip and the black "Sunsilk" tube with blue accents), it defaults to a generic hair-tutorial script and therefore selects option B rather than the ground-truth option C, which more faithfully reflects the true future frames.

Question: Based on the given video, predict future events following the observed steps. Two key future moments are provided: one where the person is seen picking up a large bottle of Ciroc Apple vodka and pouring it into the Smirnoff Ice bottle with the decorative candy strip still on it, and later, a scene where additional Ciroc Apple vodka is poured into the bottle, followed by a red sauce from a yellow bottle. Which intermediate event is most likely to occur between these two moments?

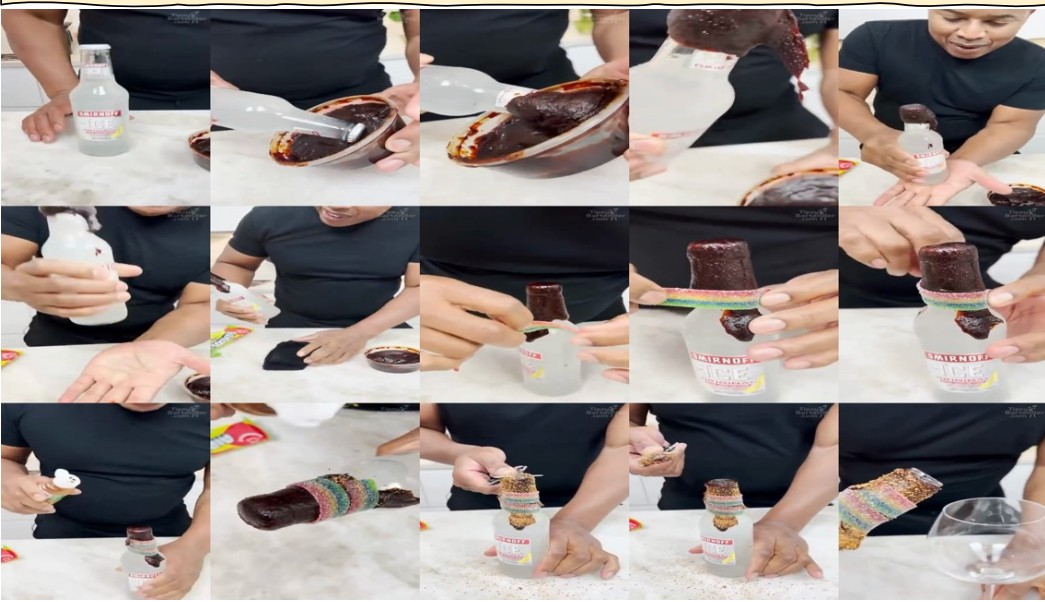

**A: The person adds chopped fruits, including pieces of apple and strawberry, into the bottle.**

B: The person shakes the bottle and places it on a black cloth.

C: The person wraps a rainbow-colored candy strip around the neck of the bottle.

D: The person uses a wooden skewer to assemble a garnish made of various colorful gummy candies.

Figure 15: The visualization of FutureBench ID: `wwNaAxypK4o`

**NEP Trained Model Output for Task in Figure 15**

Let's break down the video step by step to predict what might happen next. The person starts by preparing a Smirnoff Ice bottle, coating it with dark brown liquid (likely melted chocolate) using a spoon. Then they dip the bottle into a bowl of white powder (probably sugar or salt), creating a textured coating. After that, they wrap a colorful rainbow candy strip around the bottle, securing it with a toothpick. \n\nNext, they place the decorated bottle on a white surface and pour clear liquid (Smirnoff Ice) into a glass with ice cubes. Then there's a transition to them holding a large Ciroc Apple bottle, pouring its contents into the Smirnoff Ice bottle. The process continues with adding more Ciroc Apple and a red sauce from a yellow bottle.\n\nNow, considering the sequence, the person is likely creating a layered cocktail or mixed drink. The decorative elements suggest a focus on presentation, possibly for a social media tutorial or a creative recipe showcase. The use of different vodkas (Smirnoff Ice and Ciroc Apple) and the addition of a red sauce (maybe something like strawberry syrup or hot sauce?) indicates mixing flavors.\n\nAfter pouring the Ciroc and the red sauce into the Smirnoff Ice bottle, the next logical step would be to seal or close the bottle. Since they used a toothpick to secure the candy strip, maybe they'll add another element to keep it closed, like a lid or a decorative cap. Alternatively, they might shake the bottle to mix the ingredients, which would be a common step in cocktail making. However, since the bottle is already filled with layers, shaking could mix everything, but if they want to preserve the layers, they might not shake it. Instead, they could present the finished product, perhaps placing it back on the counter or adding a garnish on top.\n\nLooking at the previous steps involving meticulous decoration (chocolate coating, sugar/salt, candy strip), the next action might involve adding a final touch, like sprinkling more toppings, drizzling something else, or inserting a skewer with colorful candies as a garnish. The presence of the red sauce suggests a spicy or fruity element, so maybe they'll add a complementary garnish, like a strawberry slice or apple piece, especially since Ciroc Apple is mentioned.

---

**NEP Trained Model Output for Task in Figure 15 (Cont.)**

Alternatively, they might use the wooden skewer again to insert a gummy candy or other decorative item.\n\nAnother possibility is that after filling the bottle, they might demonstrate pouring the mixture into glasses, but since the current scene shows pouring into a glass with ice earlier, maybe they're building a layered drink inside the Smirnoff Ice bottle itself. If that's the case, the next step could be adding another ingredient on top, like a foam or another liquid layer, but the video doesn't mention other liquids beyond the Ciroc and red sauce. Alternatively, they might present the finished bottle as a centerpiece, maybe placing it in a visually appealing setup with other drinks or snacks.\n\nConsidering the transitions in the video described, each step builds on the previous one with increasing decoration and mixing. The final steps would likely emphasize presentation, so the person might place the bottle on a tray with other items, add a straw or a decorative pick, or even take a sip to taste. However, since the last action was pouring the red sauce, the immediate next step could be sealing the bottle, perhaps using the toothpick again to attach a lid or a small umbrella, or maybe adding a final sprinkle of the white powder (like powdered sugar) for visual contrast.\n\nGiven that the person has been meticulous in their decorations, the prediction should align with continuing that trend. Therefore, the most probable next actions are either sealing the bottle with a decorative element or adding a final garnish. Since they've already secured the candy strip with a toothpick, maybe they'll add another layer of decoration on top, such as drizzling chocolate or sprinkling herbs, but the red sauce might be part of the flavor profile rather than decoration. Alternatively, they might present the bottle as completed, perhaps placing it in a display or adding a label with the recipe name.\n\nAnother angle is considering the tools used: they have a wooden skewer, toothpick, and bowls for coating. They might use the skewer again to insert more candies or create a stand for the bottle. But since the bottle is upright and filled, maybe they'll add a skewer through the cap to hold a garnish.\n\nIn summary, based on the progression from decorating the bottle to adding multiple liquids, the next logical step is either sealing the bottle with a final touch or presenting the finished product. Given the prior use of toothpicks and candy strips, adding a final decorative element seems plausible. So, the prediction would be that the person will secure the bottle with a lid or a decorative element, possibly using the toothpick again, and then present the finished drink, maybe placing it alongside other items or demonstrating consumption.\nThe answer is B.

**Question:** Based on the given video, predict future events and fill in the potential events in the following future sequence: 1. [?] 2. [?] 3. The video concludes with the person adjusting their hair, smoothing it down, and showcasing its final smooth and shiny texture. Which option best describes the intermediate events that occur after the observed scenes?

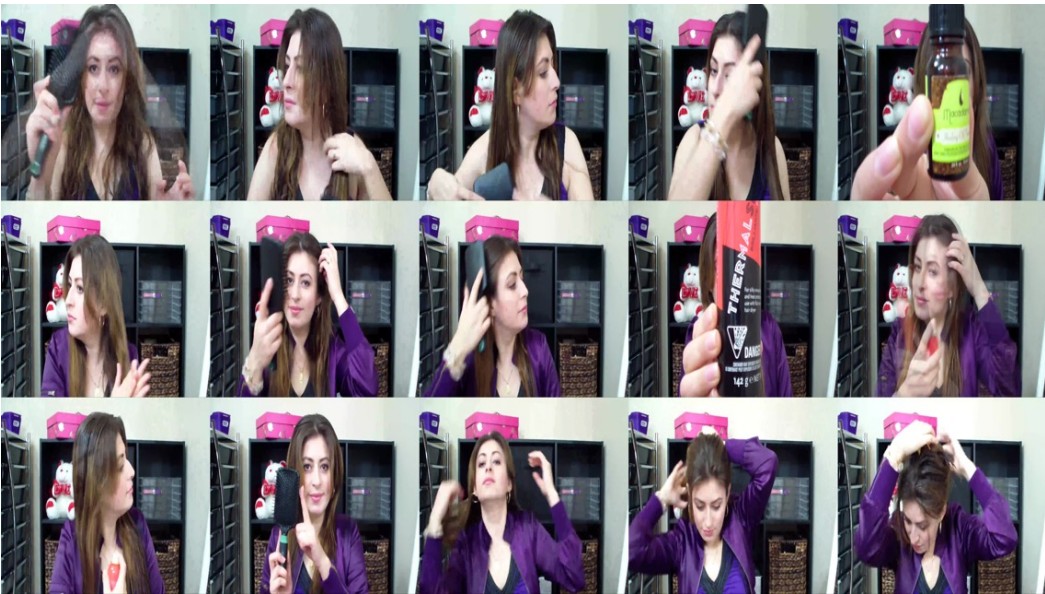

A: 1. The person gathers their hair into a ponytail and styles it using a black hair dryer with a white round brush; 2. The person applies an extra coat of product to the hair.

B: 1. The person sectioned the hair using additional clips and re-applied Moroccan Healing Oil; 2. The person uses the red spray bottle for final touches on the hair.",

**C: 1. The person gathers their hair into a ponytail using a yellow hair clip and dries and styles it with a black hair dryer and a white round brush; 2. The person holds up a black tube labeled 'Sunsilk' with blue accents, examines it, and then puts it down.**

D: 1. The person repositions the white teddy bear on the shelf while spattering thermal spray; 2. The person uses a new technique of running fingers through the hair, creating loose curls.

Figure 16: The visualization of FutureBench ID: `v_RkCSfz8GeNg`

**NEP Trained Model Output for Task in Figure 16**

Let's break down the video step by step to predict what comes next. The person starts by showing their hair routine, using specific products like the Moroccan Oil and Thermal Spray. They sectioned their hair, applied products, used a brush, and then clipped their hair up. The key actions here are methodical application of each product, sectioning, and styling.\n\nAfter clipping their hair up, the next logical step would be to either continue drying another section or apply more products. Since they already used the Thermal Spray and the black tube (possibly a heat protectant), maybe they'll move to another part of the hair. The mention of gathering into a ponytail suggests they might be preparing to style it further. The presence of the hair dryer and brushes indicates they might start blow-drying the clipped sections, perhaps using the round brush for volume.\n\nAlso, since they showed the products to the camera, they might demonstrate the application on the clipped sections, maybe applying more oil or thermal spray. The final steps would likely involve using the hair dryer to dry the clipped hair, then brushing it out for volume. The ponytail might be styled into a sleek look or incorporated into the routine for added texture.\n\nConsidering the flow, after clipping, they might take a section from the ponytail area, apply product, and use the brush and dryer to smooth or add shine. Then repeat with other sections until all hair is styled. The conclusion would show the final look, as mentioned in the caption. So the intermediate steps would involve continuing the process on the remaining hair, possibly using the same products again, and then finishing with the dryer and brush to achieve the smooth texture shown at the end.\n\nLooking at the options provided, the one that aligns with these steps is B. It mentions sectioning with clips, reapplying oil, and using the spray bottle for final touches. That fits the flow. Option A has them gathering into a ponytail first, which might come later, but the current step is clipping. Option C involves a different clip color and a different product label, which doesn't match. Option D talks about repositioning the teddy bear, which seems unrelated. So B is the most plausible.\n\nTherefore, the prediction should outline the continuation of styling the clipped sections, applying products, using the tools shown, leading to the final smooth result. The answer is B.\nThe answer is B.

## C.4 THREE TYPES OF LOGIC REASONING IN VIDEO INSTRUCTION TUNING

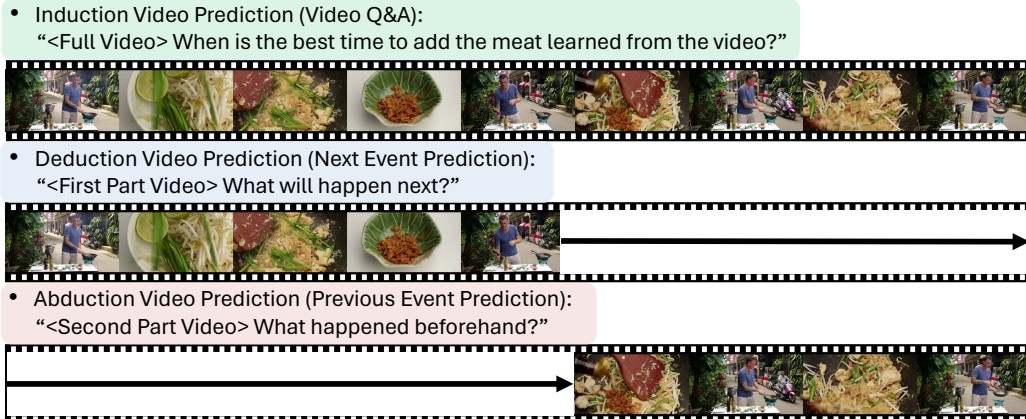

Figure 17: **Three types of logic reasoning in video instruction tuning tasks**. (1) Induction (Video Q&A): The model watches entire video sequences and learns common event patterns and temporal relationships, building an internal "engine" of how visual events unfold over time. (2) Deduction (**Next Event Prediction**): Given the first part of a video, the model uses its learned causal and commonsense knowledge to extrapolate and predict the most likely next events. (3) Abduction (Previous Event Prediction): Presented with the final segment of a video, the model reasons backward to hypothesize plausible prior events or hidden causes that explain the observed outcome.

## D  TRAINING STRATEGY

**Supervised Fine Tuning (SFT).**    We fine-tune the MLLM on V1-33K using standard supervised learning. The model receives the first segment of a video caption and predicts its continuation, training via cross-entropy loss. This stage instills basic predictive capabilities, allowing the model to directly imitate ground-truth future event descriptions.

**Critique Fine Tuning (CFT).**    CFT is a strategy where models learn to critique noisy responses instead of simply imitate answers Wang et al. (2025). We leverage critique data generated by an external LLM (e.g., GPT-4) that identify strengths and errors in model predictions relative to ground-truth continuations. During fine-tuning, the model learns to refine flawed continuations or evaluate predictions based on provided critiques, internalizing feedback to enhance logical consistency and predictive accuracy.

**Distillation Tuning (Distill).**    We employ knowledge distillation from DeepSeek-R1, a strong reasoning model. For each sample, DeepSeek-R1 generates detailed reasoning steps and a predicted caption. The student model is fine-tuned to reproduce this entire reasoning sequence, adopting structured inferential patterns to improve both reasoning and prediction accuracy.

**Mix Tuning (Mix).**    We combine SFT, CFT, and Distillation methods equally in each training epoch. By interleaving direct predictions, critique-informed refinements, and explicit reasoning demonstrations, the model integrates various supervision signals. This mixed strategy promotes robust learning, balancing factual accuracy, critical feedback integration, and structured reasoning capabilities.

## E  PROMPTS

### E.1  EVENT IDENTIFICATION PROMPT

This prompt ensures structured extraction of discrete events from raw captions.

**Event Identification**

```
Below is the video caption:
{video_caption}

Task:
1. Identify and list the events (scenes) in the video in sequential
    order (e.g., Scene 1, Scene 2, etc.).
2. For each scene, provide a description.

Please return your answer in a valid JSON format exactly as follows
    (with no extra text):

{
  "events": [
    {"scene": "Scene 1",
     "description": "Brief description of scene 1"},
    {"scene": "Scene 2",
     "description": "Brief description of scene 2"},
    ...
  ]
}
```

## E.2 CAUSAL ANALYSIS AND SPLITTING SUITABILITY PROMPT

This prompt analyzes the causal dynamics among events and determines an optimal split point that yields a stronger supervision signal for future prediction.

**Causal Analysis and Splitting Suitability Prompt**

```
Below are the extracted events from the video:
{json.dumps(event_identification_result, indent=2)}

Original video caption:
{video_caption}

Task:
1. Analyze the causal relationships among these events.
2. Determine whether the video is suitable to be split into two parts
    for causal inference (i.e., given the first part, can we predict
    what happens in the second part?).
3. If it is suitable, specify the optimal split point (for example,
    'between Scene A and Scene B').

Please provide your answer in a valid JSON format exactly as follows
    (with no extra text):

{
  "suitable": "yes" or "no",
  "optimal_split_point":
    "between Scene X and Scene Y",
  "reasoning":
    "Detailed explanation of the causal relationships
     and the split decision."
}
```

### E.3 CAPTION SPLITTING PROMPT

This prompt divides the caption into meaningful segments at the identified split.

---

**Caption Splitting Prompt**

```
Using the identified events and the optimal split point, split the
original video caption into two parts. The optimal split point is
given in the format 'between Scene X and Scene Y'. This means that
all scenes up to and including Scene X should be included in the
first part ('caption_part1'), and all scenes from Scene Y onward
should be included in the second part ('caption_part2').

The identified events:
{json.dumps(event_identification_result, indent=2)}

and the optimal split point:
{casual_analysis_result["optimal_split_point"]}

Original video caption:

{video_caption}

Return your answer in a valid JSON format exactly as follows (no
extra text):

{
  "caption_part1": "Text for first part",
  "caption_part2": "Text for second part"
}
```

---

### E.4 CHAIN-OF-THOUGHT REASONING & FUTURE PREDICTION PROMPT

This prompt guides the reasoning model to forecast upcoming events, and the resulting reasoning trace is then used for training.

---

**Chain-of-Thought Reasoning & Future Prediction Prompt**

```
You have advanced visual perception abilities and can analyze videos
as if you are watching them in real time. You will be provided with a
detailed description of a video (caption). Interpret this description
as if it represents your actual dynamic visual experience rather than
just text.

Based on the scene, analyze and predict future events. Provide
concise, natural, and confident prediction about the video's future
events. Speak as if you are directly observing the events, avoiding
any reference to reading text or captions. If details are ambiguous,
express natural uncertainty (e.g., "It appears that ...").

Caption:

{caption_part1}
```

---

### E.5 FUTURE PREDICTION CRITIQUE PROMPT

This prompt critically evaluates the alignment of predictions with the actual video outcome.

**Future Prediction Critique Prompt**

```
Task:
Review the caption of the second part of a video as the ground truth
and evaluate whether the future prediction (derived from the first
part of the video) aligns with the actual events.

What actually happened in the second part of the video:

{caption_part2}

Prediction (derived from the first part of the video):

{prediction_content}

Reasoning behind the prediction:

{reasoning_content}

Instructions:
1. Analyze the prediction and the reasoning provided, considering how
well they align with the ground truth.
2. Note that accurately predicting future events is inherently
challenging; allow for minor discrepancies and avoid overly strict
judgments.
3. Think step by step and provide a critique of the prediction and
its underlying reasoning.
4. Conclude your analysis by stating either "Conclusion: right" if
the prediction aligns well, or "Conclusion: wrong" if it does not.

Output:
Return your analysis in a valid JSON format exactly as shown below
(do not include any extra text):

{
  "Critique":
    "Your critique of the prediction and its underlying reasoning",
  "Conclusion": "right"/"wrong"
}
```

## E.6 REWRITE PROMPT

The rewriting prompt serves as a data-cleaning step to further refine the wording of the generated reasoning traces.

---

**Rewrite Reasoning Prompt**

```
You will receive a snippet of text that references a "description" or
"caption" of a video. Your task is to produce a **nearly identical**
version of that text with **minimal** changes, focusing on the
following:

1. **Replace references to "description" or "caption"** with wording
that references **"the video."**
    – For example, "The description says..." could become
      "The video shows..."
    – "The caption suggests..." could become
      "The video suggests..."
    – Make sure the replacement sounds natural but does
      **not** otherwise change the meaning.

2. **Preserve all line breaks, punctuation, and spacing** as much as
possible, and make **no additional edits** outside of these
replacements.

3. You should only output the rewritten content.

Here is the input:
{reasoning_content}
```

---

**Rewrite Prediction Prompt**

```
You will receive a snippet of text that references a "description" or
"caption" of a video. Your task is to produce a **nearly identical**
version of that text with **minimal** changes, focusing on the
following:

1. **Replace references to "description" or "caption"** with wording
that references **"the video."**
    – For example, "The description says..." could become
      "The video shows..."
    – "The caption suggests..." could become
      "The video suggests..."
    – Make sure the replacement sounds natural but does
      **not** otherwise change the meaning.

Here is the input:
{prediction_content}
```

## E.7 FUTUREBENCH CONSTRUCTION PROMPT

**FutureBench 1-Hop Question Construction Prompt**
*This prompt aims to generate the 1-hop QA pairs of FutureBench.*

---

**FutureBench 1-Hop Question Construction Prompt**

```
You are an expert in video understanding. Your task is to generate
one multiple-choice question to assess the video understanding
ability of a test model. You are given the meta information about a
video that includes:
- Video captions: A complete description of the entire video for your
reference.
- Scene descriptions: Detailed descriptions of key scenes throughout
the video.
- Observed Scenes: Scenes in the given video that the test model can
observe.
- Last Scene: The last scene of the entire video.

Requirements:

1. Question Content:
- Given the video with observed scenes (scene 1 to k), the question
should force the test model to predict future events (scene k+1 to
scene n) and ask what intermediate events would be supposing scene n
is given and scene n is the potential future end.
- For example, "Question": "Based on the given video, predict future
events and fill in the potential events in the given future events:
1. [?] 2. [describe scene n]. "Options": A/B/C/D. [describe scene for
slot 1]
- Keep the event slot [?] to be filled.
- Construct the future event gap so that it is hard enough. For
example, wrong answers could present the wrong order of the predicted
future events.
- Avoid using scene id in the question and start the question from
"Based on the given video, ..."

2. Question Format:
- Create one multiple-choice question with four answer options: A, B,
C, and D.
- Ensure only one correct answer and that the remaining three options
are wrong.
- Only output required question-answer pairs shown in the output
structure.

Output structure:

{output_structure}

Please generate an example question based on the following input data.

Input Data:
- Video captions: {caption}
- Scene descriptions: {event}
- Observed Scenes: {obs}
- Last Scene: {last}
```

**FutureBench 2-Hop Question Construction Prompt**
*This prompt aims to generate the 2-hop QA pairs of FutureBench.*

---

**FutureBench 2-Hop Question Construction Prompt**

You are an expert in video understanding. Your task is to generate
one multiple-choice question to assess the video understanding
ability of a test model. You are given the meta information about a
video that includes:
- Video captions: A complete description of the entire video for your
reference.
- Scene descriptions: Detailed descriptions of key scenes throughout
the video.
- Observed Scenes: Scenes in the given video that the test model can
observe.
- Last Scene: The last scene of the entire video.

Requirements:

1. Question Content:
- Given the video with observed scenes (scene 1 to k), the question
should force the test model to predict future events (scene k+1 to
scene n) and ask what intermediate events would be supposing scene n
is given and scene n is the potential future end.
- For example, "Question": "Based on the given video, predict future
events and fill in the potential events in the given future events:
1. [?] 2. [?] 3. [describe scene n]. "Options": A/B/C/D. [describe
scene for slot 1], [describe scene for slot 2]
- Keep the event slot [?] to be filled.
- Construct the future event gap so that it is hard enough. For
example, wrong answers could present the wrong order of the predicted
future events.
- Avoid using scene id in the question and start the question from
"Based on the given video, ..."

2. Question Format:
- Create one multiple-choice question with four answer options: A, B,
C, and D.
- Ensure only one correct answer and that the remaining three options
are wrong.
- Only output required question-answer pairs shown in the output
structure.

Output structure:

{output_structure}

Please generate an example question based on the following input data.

Input Data:
- Video captions: {caption}
- Scene descriptions: {event}
- Observed Scenes: {obs}
- Last Scene: {last}

**FutureBench 3-Hop Question Construction Prompt**
*This prompt aims to generate the 3-hop QA pairs of FutureBench.*

---

**FutureBench 3-Hop Question Construction Prompt**

```
You are an expert in video understanding. Your task is to generate
one multiple-choice question to assess the video understanding
ability of a test model. You are given the meta information about a
video that includes:
- Video captions: A complete description of the entire video for your
reference.
- Scene descriptions: Detailed descriptions of key scenes throughout
the video.
- Observed Scenes: Scenes in the given video that the test model can
observe.
- Last Scene: The last scene of the entire video.

Requirements:

1. Question Content:
- Given the video with observed scenes (scene 1 to k), the question
should force the test model to predict future events (scene k+1 to
scene n) and ask what intermediate events would be supposing scene n
is given and scene n is the potential future end.
- For example, "Question": "Based on the given video, predict future
events and fill in the potential events in the given future events:
1. [?] 2. [?] 3. [?] 4. [describe scene n]. "Options": A/B/C/D.
[describe scene for slot 1], [describe scene for slot 2] [describe
scene for slot 3]
- Keep the event slot [?] to be filled.
- Construct the future event gap so that it is hard enough. For
example, wrong answers could present the wrong order of the predicted
future events.
- Avoid using scene id in the question and start the question from
"Based on the given video, ..."

2. Question Format:
- Create one multiple-choice question with four answer options: A, B,
C, and D.
- Ensure only one correct answer and that the remaining three options
are wrong.
- Only output required question-answer pairs shown in the output
structure.

Output structure:

{output_structure}

Please generate an example question based on the following input data.

Input Data:
- Video captions: {caption}
- Scene descriptions: {event}
- Observed Scenes: {obs}
- Last Scene: {last}
```

**FutureBench Interpolation Question Construction Prompt**

You are an expert in video understanding. Your task is to generate one multiple-choice question to assess the video understanding ability of a test model. You are given the meta information about a video that includes:
- Video captions: A complete description of the entire video for your reference.
- Scene descriptions: Detailed descriptions of key scenes throughout the video.
- Observed Scenes: Scenes in the given video that the test model can observe.
- Last Scene: The last scene of the entire video.

Requirements:

1. Question Content:
- Given the video with observed scenes (scene 1 to k), the question should force the test model to predict future events (scene k+1 to scene n) and ask what intermediate events would be supposing (scene k+i and scene k+j are given, k+i and k+j are potential future events).
- For example, "Question": "Based on the given video, predict future events and fill in the potential events in the given future events: 1. [describe scene k+1] 2. [?] 3. [describe scene k+i] 4. [?] 5. [describe scene k+j]. "Options": A) [describe scene k+2], [describe scene k+j-1] B) [describe scene k+j-1], [describe scene k+2] C) [describe scene k+i+1], [describe scene k+i-1] D) [describe scene k+i-1], [describe scene k+2]
- Formulate the question so that the test model would not be able to deduce the correct answer without the observed scenes.
- Formulate the question so that it is hard enough and the test model would not be able to deduce the correct answer with only commonsense knowledge.
- Avoid using scene id in the question and start the question from "Based on the given video, ..."

2. Question Format:
- Create one multiple-choice question with four answer options: A, B, C, and D.
- Answer options should be built upon the scenes after the observed scenes and before the last scene.
- Ensure only one correct answer and that the remaining three options are wrong.
- Ensure each wrong answer contains related information to the observed scene but include missing details or only part of them are correct.
- Only output required question-answer pairs shown in the output structure.

Output structure:
{output_structure}

Please generate an example question based on the following input data.

Input Data:
- Video captions: {caption}
- Scene descriptions: {event}
- Observed Scenes: {obs}
- Last Scene: {last}

## F  IMPLEMENTATION DETAILS

We conducted our experiments using two open-source frameworks: *LLaMA-Factory* Zheng et al. (2024) for supervised video instruction tuning, and *EasyR1* Zheng et al. (2025) (based on the Verl framework Sheng et al. (2024)), optimized for reinforcement learning with multimodal data.

For supervised video instruction tuning, we trained our Qwen2.5-3B-VL-Instruct and Qwen2.5-7B-VL-Instruct models using LLaMA-Factory. Both models were fine-tuned for three epochs on 8 NVIDIA A100 GPUs, employing the AdamW optimizer with a cosine learning rate scheduler, an initial learning rate of $1 \times 10^{-5}$, and a warm-up ratio of 0.1 to ensure stable training dynamics. To optimize memory usage and address computational constraints, each GPU processed one training sample per step, with gradient accumulation every two steps, effectively simulating a larger total batch size of 16 (2 steps × 8 GPUs).

For the reinforcement learning phase, experiments were executed using EasyR1 to further enhance model capabilities through multimodal refinement learning with Group Relative Policy Optimization (GRPO) Guo et al. (2025); Shao et al. (2024), also utilizing 8 NVIDIA A100 GPUs. We fine-tuned the Qwen2.5-VL-7B-Instruct model with a maximum prompt length of 4096 tokens and a response length capped at 2048 tokens. Training utilized global batch sizes of 16 samples per rollout, with micro-batches of four samples per GPU during parameter updates and eight per GPU for experience collection. We set the entropy coefficient to $1 \times 10^{-3}$ to encourage exploration and the KL-divergence loss coefficient to $1 \times 10^{-2}$ to maintain stable policy updates. Rollouts were configured to run eight steps without tensor parallelism or chunked prefill, ensuring efficient training and stable convergence. Evaluation logging was performed periodically, capturing ten generations per validation. Model checkpoints were systematically saved every 200 training iterations for comprehensive monitoring.

To accommodate varied visual inputs, in both settings, image resolutions were constrained between a minimum of 3136 pixels and a maximum of 1,605,632 pixels, ensuring consistency and computational manageability across diverse multimodal data.

For evaluation, we leveraged the open-source multimodal evaluation framework *lmms-eval* Zhang et al. (2024a). The framework encompasses all the benchmarks except SeedBench-R1 and our proposed FutureBench. To enhance reproducibility and usability, we integrated both SeedBench-R1 and FutureBench into the lmms-eval framework. Hyperparameters followed the default settings provided by lmms-eval. Specifically, we report the average score of all subtasks in TemporalBench, including the long and short QA (both binary and multiple-choice), and the short caption. For TempCompass, we report the average score of multi-choice QA, Yes/No QA, and Caption Matching. For SeedBench-R1, we report the average score of all three levels.

## G  LIMITATION AND FUTURE WORK

Despite demonstrating the effectiveness of Next-Event Prediction (NEP) in advancing temporal reasoning capabilities in Multimodal Large Language Models (MLLMs), our current work has several limitations that invite further exploration. First, NEP primarily relies on automatically generated textual descriptions for future video segments as supervision signals. Although this approach offers scalability and avoids costly human annotations, the quality of generated captions might not match human-level precision and may reflect biases inherent in the annotation models used (e.g., GPT-4o Hurst et al. (2024)). Future research could explore integrating annotations from diverse sources, such as human annotators or alternative advanced models like Gemini Team (2025), to enhance annotation quality and reduce biases.

Second, while our proposed V1-33K dataset encompasses diverse scenarios, it may not fully capture all possible real-world video contexts, particularly highly specialized or infrequent event sequences. Extending this dataset by including additional domains, incorporating larger datasets, or employing synthetic video generation techniques could further enhance the diversity and robustness of the dataset, thereby strengthening models' temporal reasoning abilities. In particular, our current NEP data are constructed from the *intersection* of video corpora that already support captioning and QA-style tasks in order to enable a fair task-level comparison. This design choice reduces the effective diversity of video sources and, together with the limited set of temporal benchmarks we evaluate on, may

contribute to the observed saturation of downstream performance when scaling NEP data beyond 5K samples.

Third, current state-of-the-art (SOTA) models often integrate diverse instruction-tuning datasets and tasks and leverage model merging strategies to optimize performance across benchmarks. By contrast, our experimental protocol primarily focuses on comparing different tasks individually and deliberately trains NEP in isolation, without combining it with QA or captioning data. Since most evaluation benchmarks adopt a QA-style format, this setup places NEP at a slight disadvantage and may understate its potential benefits in more realistic multi-task training recipes. Future research aimed at achieving SOTA performance across a wider array of benchmarks could benefit from (i) collecting more diverse video sources, (ii) scaling NEP to larger and less restricted corpora, and (iii) exploring combined instruction-tuning strategies and model merging approaches that jointly optimize NEP with QA and captioning objectives under a carefully designed curriculum.

Addressing these limitations will significantly enhance the reliability, generalizability, and depth of temporal reasoning capabilities in video-based multimodal language models.

## H   BROADER IMPACTS

The proposed next-event prediction task has the potential to have a significant positive societal impact by improving multimodal models' temporal reasoning capabilities, increasing their utility in applications such as video-based surveillance, assistive technology, and educational content generation. Improved predictive understanding of dynamic events could also help in safety-critical situations like traffic management and emergency response systems. However, there are some drawbacks, such as the risk of reinforcing biases embedded in training datasets, which is exacerbated by the reliance on automatically generated captions without human oversight. Careful consideration, transparent documentation, and strict ethical oversight will be essential to mitigate these risks and ensure responsible deployment.

## I   LICENSES

We use standard licenses from the community. We include the following licenses for the codes, datasets and models we used in this paper.

Datasets & Benchmarks:

- VideoMME (Fu et al., 2024): CC BY-NC 4.0
- MVBench (Li et al., 2024d): MIT
- LongVideoBench (Wu et al., 2024): CC-BY-NC-SA 4.0
- TemporalBench (Cai et al., 2024): MIT
- TempCompass (Liu et al., 2024b): CC BY-NC 4.0
- SeedBench-R1 (Chen et al., 2025): Apache License 2.0
- LLaVA-Video-178K (Zhang et al., 2023): Apache License 2.0

Codes:

- verl (Sheng et al., 2024): Apache License 2.0
- EasyR1 (Zheng et al., 2025): Apache License 2.0
- LLaMA-Factory Zheng et al. (2024): Apache License 2.0

Models:

- Qwen2.5-VL-7B-Instruct (Bai et al., 2025): Apache License 2.0
- Qwen2.5-VL-3B-Instruct (Bai et al., 2025): Apache License 2.0
- OpenAI API (Hurst et al., 2024): OpenAI API Terms of Use

## J    LLM USAGE

We used an OpenAI LLM (GPT-5) as a writing and formatting assistant. In particular, it helped refine grammar and phrasing, improve clarity, and suggest edits to figure/table captions and layout (e.g., column alignment, caption length, placement). The LLM did not contribute to research ideation, experimental design, implementation, data analysis, or technical content beyond surface-level edits. All outputs were reviewed and edited by the authors, who take full responsibility for the final text and visuals.

