# OpenReview forum: "Fostering Video Reasoning via Next-Event Prediction"
_ICLR.cc/2026/Conference — ICLR 2026 Poster_

### Official Review · Reviewer_YKsb · 2025-10-24

**Soundness:** 2
**Presentation:** 3
**Contribution:** 2
**Rating:** 4
**Confidence:** 4

**Summary:**

This paper proposes "next-event prediction" (NEP), a self-supervised learning task designed to teach multimodal models temporal reasoning from video. NEP trains models to predict future events by observing past video frames. To support this, the authors introduce the V1-33K dataset and the FutureBench benchmark for evaluation. Experiments show that NEP is a scalable and effective method for fostering temporal reasoning in MLLMs.

**Strengths:**

The intuition of the paper is straightforward and promising, to train a model that predicts future events to improve their performance on temporal benchmarks, rather than just doing QA or captioning, which are either too finegrained or too general in the temporal sense, does not help the model learn to predict based on observation. The authors conducted very extensive experiments, such as different kinds of training strategies (SFT, distillation, RL), and introduced FutureBench, one of the first benchmarks that incorporates multi-hop extra/intrapolation QAs for temporal reasoning.

**Weaknesses:**

1. The most important point I want clarification on is the soundness of the pipeline for curating training data. As the authors have even mentioned themselves, it is difficult to predict the future solely based on the past events, simply because there are a lot of factors not represented in the video frames. In Figure 1, the authors' example of basketball perfectly describes this difficulty. Yet, in line 215 and the *future prediction verification prompt* in the appendix, the authors only stated that they would "allow some variation of difficulty" or "allow for minor discrepancies and avoid overly strict judgments". These instructions and descriptions are not only vague but also brings questions to the soundness of the data curation pipeline. In this case, wouldn't the "players may reduce tempo", "coach calls for timeout", "players attack even more aggressively" and many more events in the basketball example all be equally reasonable? In this case, how can one assure, since there is only one ground truth, that the "minor discrepancies" allowed for can even encompass this wide range of events possible?

2. GRPO performance only reported on FB but not other temporal benchmarks. I'm asking for this especially because FB is in-domain, which explains the huge performance gains, but the model suffered a 2-point loss on general benchmark performance.

3. Outside of the evaluated temporal benchmarks, if time permits, the authors should also try out Vinoground [1]'s video score that contains multiple segments in one video, and Tomato [2], which are also temporal benchmarks.

4. It would be better if the authors can include some data analysis on FutureBench, such as the distributions for video length, how many are 1-hop, how many are 2-hop, etc.

[1] Zhang et al, 2024, Vinoground: Scrutinizing LMMs over Dense Temporal Reasoning with Short Videos

[2] Shangguan et al, 2025, TOMATO: Assessing Visual Temporal Reasoning Capabilities in Multimodal Foundation Models

**Questions:**

See Weaknesses.

---

> ### Author Response · Authors · 2025-11-26
> **Rebuttal by Authors [1/2]**
>
> We appreciate the reviewer's thoughtful feedback. We are encouraged by their description of our intuition as "straightforward and promising" and their acknowledgment of our extensive experiments. Below we respond to the comments in **Weaknesses (W)**.
>
> ---
>
> ***W1: The most important point I want clarification on is the soundness of the pipeline for curating training data. As the authors have even mentioned themselves, it is difficult to predict the future solely based on the past events, simply because there are a lot of factors not represented in the video frames. In Figure 1, the authors' example of basketball perfectly describes this difficulty. Yet, in line 215 and the future prediction verification prompt in the appendix, the authors only stated that they would "allow some variation of difficulty" or "allow for minor discrepancies and avoid overly strict judgments". These instructions and descriptions are not only vague but also brings questions to the soundness of the data curation pipeline. In this case, wouldn't the "players may reduce tempo", "coach calls for timeout", "players attack even more aggressively" and many more events in the basketball example all be equally reasonable? In this case, how can one assure, since there is only one ground truth, that the "minor discrepancies" allowed for can even encompass this wide range of events possible?***
>
>
>
> We thank the reviewer for raising this concern. An NEP instance is defined by one video and its realized continuation: the target text describes what actually happened in the future segment of that particular video. In this sense, NEP is analogous to next-token prediction in language modeling: given a prefix, many continuations are in principle reasonable, but the model is trained on the single continuation that appears in the data, and across many such samples it learns a *conditional distribution* over futures.
>
>
> To make this analogy concrete, consider training a language model on an open-domain narrative corpus (e.g., novels or stories). Given a prefix such as "She stood at the edge of the forest, gripping the letter in her hand as the sun began to set..." (the Chapter 1st of a story), there are many continuations that are a priori reasonable (she decides to enter the forest, she turns back to the village, someone appears behind her, she drops the letter and runs, etc.). The corpus, however, only contains the single continuation that happened to be written in that particular story. We do not assume that this observed continuation is the only logically valid one; rather, we treat it as one realized sample from a broader set of plausible futures, and **this ambiguity does not diminish the value of such textual data—it is exactly how language models are routinely trained in practice**. Just as we do not require a unique logically correct continuation in the language case, in our NEP setting we acknowledge that multiple futures could be reasonable in principle, but for each training example we simply supervise on the specific future that actually occurred in that video.
>
>
> Thus, the soundness of the pipeline does not rely on the future being deterministic given the past; it only requires that each example has one coherent realized continuation and that our judge tolerates natural paraphrases and small variations around that continuation. We will clarify this point in the revised manuscript.
>
>
> ---
>
>
>
> ***W2: GRPO performance only reported on FB but not other temporal benchmarks. I'm asking for this especially because FB is in-domain, which explains the huge performance gains, but the model suffered a 2-point loss on general benchmark performance.***
>
>
>
> This is a good catch! Our intention in that section was actually to echo the same phenomenon reported in concurrent work. In our experiments, directly applying GRPO to the video LLM boosts in-domain performance on FB but causes an 2% drop on general benchmarks. This leads to our highlight in the manuscript: "SFT remains a simple yet efficient approach". Given that our preliminary experiments observed degradation on general benchmarks under GRPO and did not show consistent gains over NEP-SFT, we chose not to further scale up this configuration on other benchmark. Instead, we keep GRPO as an illustrative, secondary setting whose role is for discussing the trade-off, large in-domain gains at the cost of general performance, and to support the same conclusion as concurrent work: simple SFT is typically the more robust and reliable choice for video LLMs. This pattern is consistent with Tempsamp-R1 [3] (Table 3), which reports that GRPO performs worse than SFT on the Charades-STA (temporal) benchmark, and with Video-R1 [4], where directly using GRPO (Video-R1-7B-zero) can be worse than the baseline. Our results therefore echo the same conclusion, and we will clarify this positioning of GRPO and SFT in the revision.

---

> > ### Author Response · Authors · 2025-11-26
> > **Rebuttal by Authors [2/2]**
> >
> > ---
> >
> >
> > ***W3: Outside of the evaluated temporal benchmarks, if time permits, the authors should also try out Vinoground [1]'s video score that contains multiple segments in one video, and Tomato [2], which are also temporal benchmarks.***
> >
> >
> > We agree that evaluating on additional temporal benchmarks is valuable. In the revised manuscript, we have added $\\textrm{\\color{blue}Section 5.3}$, where we evaluate our models on TOMATO, a recent benchmark explicitly designed to test fine-grained visual temporal reasoning. Following the official evaluation protocol, we report accuracy decomposed by reasoning type (Rotation, Direction, Velocity & Frequency, Shape & Trend, Visual Cues, and Action Count). As shown in $\\textrm{\\color{blue}Table 3}$, NEP-trained models—especially with the Distill strategy—achieve a 2.32% absolute improvement over the baseline.
> >
> >
> > ---
> >
> >
> >
> > ***W4: It would be better if the authors can include some data analysis on FutureBench, such as the distributions for video length, how many are 1-hop, how many are 2-hop, etc. (ref to appendix)***
> >
> > A: Thanks for the great suggestion. You can find the data statistics of FutureBench in $\\textrm{\\color{blue}Table 6}$, $\\textrm{\\color{blue}Appendix B.2}$ in the revision for a reference.
> >
> >
> > ---
> >
> > ***References:***
> >
> > [1] Zhang et al, 2024, Vinoground: Scrutinizing LMMs over Dense Temporal Reasoning with Short Videos
> >
> > [2] Shangguan et al, 2025, TOMATO: Assessing Visual Temporal Reasoning Capabilities in Multimodal Foundation Models
> >
> >
> >
> > [3] Li, Yunheng, et al. "Tempsamp-r1: Effective temporal sampling with reinforcement fine-tuning for video llms." arXiv preprint arXiv:2509.18056 (2025).
> >
> > [4] Feng, Kaituo, et al. "Video-r1: Reinforcing video reasoning in mllms." arXiv preprint arXiv:2503.21776 (2025).

---

> > > ### Comment · Reviewer_YKsb · 2025-11-28
> > >
> > > I appreciate the authors' detailed feedback. I want one further clarification: In the "future prediction verification prompt", can you tell me if this is dedicated for the curation of training data V1-33K, the curation of FB, or for evaluating if the prediction from a model is correct, like LLM-as-a-judge?

---

> ### Author Response · Authors · 2025-12-01
> **Response to Reviewer’s Follow-Up Questions**
>
> Thank you for the supportive follow-up! Below we clarify the usage of the future prediction verification prompt.
>
> ---
>
> ***Q1: I appreciate the authors' detailed feedback. I want one further clarification: In the "future prediction verification prompt", can you tell me if this is dedicated for the curation of training data V1-33K, the curation of FB, or for evaluating if the prediction from a model is correct, like LLM-as-a-judge?***
>
> Thank you for asking this question. The short answer is that the future prediction verification prompt is used for the curation of the training data V1-33K, rather than for FutureBench construction. More specifically, this prompt is employed when generating critique data.
>
>
> We realized that this usage could be made clearer. In the revised version, we have refined $\\textrm{\\color{blue}Appendix B.1}$ and explicitly added step-by-step prompt references, so that future readers can more easily see which prompts are used at each stage of data construction.

---

### Official Review · Reviewer_ETi3 · 2025-10-26

**Soundness:** 3
**Presentation:** 3
**Contribution:** 2
**Rating:** 6
**Confidence:** 4

**Summary:**

This paper introduces Next-Event Prediction (NEP), a self-supervised training task designed to enhance temporal reasoning in multimodal large language models (MLLMs) for video understanding. NEP trains a model to generate captions for future video segments based on past frames. To support this task, the authors convert multiple existing video datasets into NEP format, forming a new training dataset named V1-33K. They also propose FutureBench, a benchmark for evaluating multi-hop temporal reasoning through the extrapolation and interpolation of future events. Experiments on seven video understanding benchmarks show that NEP consistently outperforms conventional training tasks, such as video question answering and captioning, on temporal reasoning, while preserving strong general video understanding capabilities. In addition, the authors compare several training strategies—supervised fine-tuning (SFT), critique fine-tuning (CFT), distillation (Distill), mixed tuning (Mix), and GRPO-based reinforcement learning—and provide practical insights into their relative effectiveness in different scenarios.

**Strengths:**

- The core idea of NEP is sound and innovative. It effectively enhances both visual perception and temporal reasoning through a self-supervised learning paradigm, which also provides a clear advantage in scalability.
- Results demonstrate that NEP yields notable improvements on temporal reasoning benchmarks, outperforming standard video QA and captioning.
- The comparisons against existing training tasks are well-designed and carefully controlled.
- The investigation of multiple training strategies provides practical insights into how NEP can be effectively applied under different scenarios.
- The paper is generally well-structured and easy to follow.

**Weaknesses:**

- Despite NEP’s theoretical scalability, the reported results show that downstream performance saturates with only 5k training samples. This suggests that the current dataset lacks sufficient diversity or scale to fully reveal the benefits of data scaling for NEP, thereby limiting its contribution to the community.
- As the authors acknowledge, predicting future events from a past video segment is inherently ambiguous, and the automatically constructed NEP dataset may therefore contain samples that are either trivial or excessively difficult. Although a caption analysis step is introduced to improve data quality, the paper does not provide empirical evidence demonstrating the effectiveness of this step.
- The paper does not provide sufficient evidence supporting the quality of FutureBench samples. The low accuracy of o4-mini alone is not enough to justify data quality. Including a human performance baseline or showcasing representative examples would better substantiate the benchmark’s validity.

**Questions:**

Please refer to the weaknesses.

---

> ### Author Response · Authors · 2025-11-26
> **Rebuttal by Authors**
>
> We appreciate the reviewer's encouraging feedback! We are pleased that they find the core idea of NEP "sound and innovative" and recognize its benefits for temporal reasoning. Below we respond to the comments in **Weaknesses (W)**.
>
> ---
>
> ***W1: Despite NEP’s theoretical scalability, the reported results show that downstream performance saturates with only 5k training samples. This suggests that the current dataset lacks sufficient diversity or scale to fully reveal the benefits of data scaling for NEP, thereby limiting its contribution to the community.***
>
> Thanks for the nice observation and good question. It is acknowledged that performance saturates with 5k training samples. However, we would like to highlight that the diversity of video sources also matter. In the current experimental setup, we use the same video source with caption, MCQA, OEQA to control the variable. We believe further data scaling should also scaling video sources beyond data size, together with improved data quality.
>
> ---
>
> ***W2: As the authors acknowledge, predicting future events from a past video segment is inherently ambiguous, and the automatically constructed NEP dataset may therefore contain samples that are either trivial or excessively difficult. Although a caption analysis step is introduced to improve data quality, the paper does not provide empirical evidence demonstrating the effectiveness of this step.***
>
> Thanks for this great question. We believe caption analysis is necessary to make the event segment, which is an important step in NEP. Without caption analysis, it makes non-reasonable video segmentation and results in random and noisy learning signals. In this case, we may not do the real NEP and this ablation cannot lead any new conclusive statements.
>
> ---
>
>
> ***W3: The paper does not provide sufficient evidence supporting the quality of FutureBench samples. The low accuracy of o4-mini alone is not enough to justify data quality. Including a human performance baseline or showcasing representative examples would better substantiate the benchmark’s validity.***
>
>
> Thanks for the great suggestion. We provide the samples from FutureBench in Appendix C.1. We agree that human performance baseline is good but it might take much longer time. We believe the performance of a strong text-only reasoning model is a good evidence, which substantiates the benchmark’s validity together with the representative examples.

---

> > ### Comment · Reviewer_ETi3 · 2025-11-28
> > **Response to Author Rebuttal**
> >
> > I appreciate the authors’ efforts in responding to my earlier comments. However, my key concerns remain insufficiently addressed:
> > - W1: While the authors discuss a potential solution, the diversity of the current dataset remain limited.
> > - W2: I am not fully convinced by the explanation regarding the absence of the ablation study. This ablation seems essential for validating the effectiveness of Caption Analysis, especially since the paper itself marks this component as optional in Section 3.2.
> > - W3: I still believe that the performance of o4-mini alone is insufficient to demonstrate the data quality. The auhors may consider sampling a subset of data to manually verify the quality.

---

> ### Author Response · Authors · 2025-12-01
> **Response to Reviewer’s Follow-Up Questions [1/2]**
>
> We thank the reviewer again for the constructive feedback and the supportive overall assessment of our work. We also appreciate the restatement of the key concerns, which has helped us further refine the manuscript!
>
> In this revised version, we not only add the *caption analysis* ablation requested in both the initial W2 and the follow-up W2, but also provide a more detailed *training data scaling study* and discussion in $\\textrm{\\color{blue}Figure 6}$ and $\\textrm{\\color{blue}Section 5.2}$ to better address the reviewer’s concerns about data diversity and the effect of scale.
>
>
> ---
>
> ***W1: Results show that downstream performance saturates with only 5K training samples. This suggests that the current dataset lacks sufficient diversity or scale to fully reveal the benefits of data scaling for NEP.***
>
> *(Please allow us to reuse the concern sentence from your initial comments as the statement of this weakness, as we believe it best captures your current concern.)*
>
> We partly agree with the reviewer that the current dataset does not fully expose NEP’s data-scaling behavior. The observed saturation phenomenon partly stems from  **1. our deliberate choice to restrict NEP to the intersection of video corpora** that already support VQA and captioning, in order to enable a fair task-level comparison. This naturally reduces source diversity and,  **2.the limitation of temporal benchmarks we initially evaluate on**, contributes to the observed saturation. Importantly, this constraint is not inherent to NEP: in a practical state-of-the-art recipe one would simply lift this restriction by collecting more diverse videos and combining NEP with QA/captioning data during training, which we now explicitly highlight as future work.
>
> Specifically, we believe the observed saturation is driven by the following four factors:
>
> 1. **NEP’s objective focuses on temporal reasoning rather than world knowledge.**
>    NEP is specifically designed to enhance temporal reasoning, not to inject additional general world knowledge. Therefore, we do not expect NEP-only training to *consistently* improve general-purpose benchmarks as the data scale grows (the first-row subfigures in Figure 5). In this regime, maintaining comparable general performance (i.e., avoiding degradation) is already a desirable outcome.
>
> 2. **Restricted video sources reduce effective data diversity.**
>    To cleanly isolate the effect of the *training task* itself, we construct NEP data from the **intersection** of video corpora that have already been used for other training tasks, including captioning, OEQA, and MCQA. Using this core set of videos allows a controlled, task-level comparison under a fixed video corpus, but it also means that NEP is trained on a restricted set of video sources. As a result, when we add more NEP samples from essentially the same narrow distribution, increasing the data size yields diminishing gains—exactly the saturation effect observed in Figure 5.
>
> 3. **Our protocol trains NEP in isolation, which is slightly disadvantaged for QA-style evaluation.**
>    Our experimental protocol deliberately trains NEP *alone* and compares it against models trained solely on MCQA or OEQA, so that we can study "training task vs. training task" under a fixed tuning budget. Because most evaluation benchmarks are formulated in QA format, *this design places NEP at a slight disadvantage*: NEP-only tuning introduces a shift between the NEP objective and the downstream QA-style usage.
>
> Note, in a practical recipe for pursuing state-of-the-art performance, one would (i) be *free* to collect more diverse video sources and (ii) combine NEP with QA and captioning data, with a carefully designed curriculum, rather than training with NEP alone. We explicitly mention this in the limitation section.
>
> 4. **Existing temporal benchmarks under-represent NEP’s benefits; TOMATO shows a more positive scaling trend.**
>    As discussed in the recent temporal reasoning benchmark TOMATO [1], many widely used temporal video benchmarks allow models to exploit shortcuts, enabling them to answer correctly using a single, a few, or even out-of-order frames. In contrast, the newly added $\\textrm{\\color{blue}Figure 6}$ shows that, on TOMATO, NEP exhibits a more positive data-scaling trend: increasing the NEP training data generally improves temporal performance (with some non-monotonic fluctuations). This suggests that NEP’s scalability is better revealed on harder temporal benchmarks and that the saturation seen in our current experiments is partly due to limitations of existing datasets.

---

> ### Author Response · Authors · 2025-12-01
> **Response to Reviewer’s Follow-Up Questions [2/2]**
>
> ---
>
>
> ***W2: I am not fully convinced by the explanation regarding the absence of the ablation study. This ablation seems essential for validating the effectiveness of Caption Analysis, especially since the paper itself marks this component as optional in Section 3.2.***
>
> We thank the reviewer for insisting on this ablation. We agree that it is important for validating the effectiveness of the Caption Analysis component. Therefore, in the revised version, we have added a dedicated ablation study in $\\textrm{\\color{blue}Section 5.2}$ ($\\textrm{\\color{blue}Table 3}$, Effect of Caption Analysis) that directly compares NEP tuning **with vs. without** Caption Analysis on the same underlying video corpus.
>
> Concretely, starting from the untuned Qwen2.5-VL-7B-Instruct baseline (T-Avg.: 49.7), applying SFT on NEP data constructed in a simple way without Caption Analysis already yields a modest improvement on temporal benchmarks (T-Avg.: 50.9) while largely preserving performance on general benchmarks. When we further enable Caption Analysis on top of the same video corpus, temporal performance improves consistently across FutureBench and other temporal benchmarks, increasing T-Avg. to 52.6. These results demonstrate that Caption Analysis is an effective step for improving the quality and challenge level of the constructed data.
>
> Regarding the “optional” wording in Section 3.2: our intention was to emphasize that the pipeline can still generate NEP data in settings where Caption Analysis (or Reasoning & Critique) is unavailable. To avoid misunderstanding for other readers, we have refined $\\textrm{\\color{blue}Section 3.2}$ in the revision to clarify this point.
>
>
> ---
>
> ***W3: I still believe that the performance of o4-mini alone is insufficient to demonstrate the data quality. The auhors may consider sampling a subset of data to manually verify the quality.***
>
> We agree that human-checking is important. As described in our initial submission $\\textrm{\\color{blue}Appendix B.3}$ (Human-in-the-loop Quality Review), we already performed manual inspection on randomly sampled FutureBench instances during construction.
>
> In addition, to give reviewers a more direct sense of the benchmark quality beyond aggregate numbers, we provide multiple FutureBench examples in $\\textrm{\\color{blue}Figures 8, 9, 10, 13, 14}$, covering diverse scenarios and difficulty levels.
>
> Taken together, **(1)** the human-in-the-loop review on sampled data, **(2)** the challenging performance of strong LMs such as o4-mini, and **(3)** the qualitative examples are intended to provide a more comprehensive and concrete demonstration of FutureBench’s quality and non-triviality.
>
>
> ---
>
> ***References:***
>
> [1] Shangguan et al, 2025, TOMATO: Assessing Visual Temporal Reasoning Capabilities in Multimodal Foundation Models

---

### Official Review · Reviewer_m8KR · 2025-10-29

**Soundness:** 3
**Presentation:** 2
**Contribution:** 2
**Rating:** 4
**Confidence:** 2

**Summary:**

The paper proposes Next-Event Prediction (NEP) as a new self-supervised learning task to enhance temporal reasoning in multimodal large language models (MLLMs). Experimental results across multiple temporal reasoning benchmarks demonstrate that NEP improves reasoning capabilities without sacrificing general video understanding performance.

**Strengths:**

- Originality: It introduces a distinct and underexplored formulation, NEP, bridging video understanding and autoregressive reasoning.

- Quality: Demonstrates consistent improvement across multiple reasoning tasks (Table 1–3).

- Clarity: The paper provides with well-structured writing and clear motivation.

- Significance: The paper shows potential to influence future temporal reasoning research and dataset design, contingent on improved empirical robustness.

**Weaknesses:**

- No qualitative visualization of predicted events or linguistic coherence.
- No error analysis or failure case study contrasting NEP and baseline models.
- No ablation on architecture or training objectives to isolate contribution of NEP loss components.

**Questions:**

- Can authors provide with qualitative examples of NEP predictions vs. ground truth to illustrate temporal understanding?
- Can authors conduct ablation studies on each NEP component?
- Can authors discuss computational cost and scalability trade-offs of NEP relative to captioning or QA fine-tuning?
- What are the typical failure modes of the NEP model when predicting future events? How does the model’s performance degrade when event boundaries are noisy or poorly segmented? Could the authors provide a qualitative analysis or visualization of failed predictions to clarify these limitations?
- How does NEP handle scenarios with multiple plausible futures?

---

> ### Author Response · Authors · 2025-11-26
> **Rebuttal by Authors [1/3]**
>
> ***W1: No qualitative visualization of predicted events or linguistic coherence.***
>
>
> We agree that illustrating concrete examples of predicted events and their linguistic coherence is direct way for understanding what NEP actually learns. In the revised version, we have added a dedicated qualitative analysis section and multiple visualizations.
>
>
> 1. **(Visualization of NEP training data)** To clarify the nature of the supervision signal and its linguistic structure, we now visualize NEP training samples in $\\textrm{\\color{blue}Appendix C}$ (new $\\textrm{\\color{blue}Figures 12}$ and $\\textrm{\\color{blue}13}$). For each example, we show: (i) the sequence of key video events from the "seen" part, and (ii) the corresponding future-event description. These visualizations highlight that our targets are high-level future events that summarize how the situation evolves, rather than short, local actions. This makes explicit what kind of future structure the model is trained to capture on the input–label side.
> 2. **(Qualitative comparison of predicted event)** We further added qualitative case studies comparing Qwen2.5-VL-7B with its NEP-trained variant ($\\textrm{\\color{blue}Appendix C.3}$, new $\\textrm{\\color{blue}Figures 10}$ and $\\textrm{\\color{blue}11}$) from the model-output side. Each case presents: (i) video frames, (ii) the candidate future-event options, and (iii) the model’s chosen option. We annotate why the NEP-trained model’s prediction is both temporally and linguistically coherent, while the base model often fails along one of these axes. For example, in the laundry-room case ($\\textrm{\\color{blue}Figure 10}$), the NEP-trained model places the “taking the mug and leaving” action in the past segment and selects a future event that continues the ongoing laundry process, maintaining entity consistency and temporal ordering. The vanilla model, by contrast, over-focuses on the mug and chooses an option that effectively re-describes the past, breaking the narrative flow. In the dog-grooming example ($\\textrm{\\color{blue}Figure 11}$), the NEP-trained model selects intermediate product-display shots that fit the ad-like storyline leading into the final exterior storefront, while the base model defaults to “more grooming scenes,” which is linguistically plausible but temporally less coherent with the visual narrative structure. These examples explicitly visualize how NEP improves both temporal alignment with the video and linguistic coherence of the predicted future events.
>
>
> Overall, the newly added visualizations show that NEP-trained models produce future-event descriptions that are not only  fluent, but also tightly aligned with the video segment and temporally coherent.
>
> ---
>
> **W2: No error analysis or failure case study contrasting NEP and baseline models.**
>
> Thank you for highlighting the need for explicit error analysis and failure-case comparison between NEP and baseline models. We agree that understanding where NEP helps and where it fails is crucial. Since this point substantially overlaps with: "*Q1: Can authors provide with qualitative examples of NEP predictions vs. ground truth to illustrate temporal understanding?*", we address them together here.
>
>
> In the revised version, we have added a qualitative error-analysis section contrasting NEP with the Qwen2.5-VL-7B ($\\textrm{\\color{blue}Appendix C.3}$). This section includes (i) cases where the NEP-trained model answers correctly while the vanilla model fails and (ii) cases where NEP still fails. For each example, we visualize video frames, list the candidate future-event options, and annotate the outputs of both models. As we discussed in the reply for W1, these examples explicitly visualize how NEP improves both temporal alignment with the video and linguistic coherence of the predicted future events.
>
>
> Taken together, these failure-case visualizations and patterns complement our quantitative results: they show not only how NEP improves temporal reasoning over the baseline, but also where it breaks down and why, thereby addressing your request for an explicit error and failure-case study contrasting NEP with the base model.

---

> > ### Author Response · Authors · 2025-11-26
> > **Rebuttal by Authors [2/3]**
> >
> > ---
> >
> > ***W3: No ablation on architecture or training objectives to isolate contribution of NEP loss components.***
> >
> > Thank you for the comment. We view this point as closely related to your question point (*Q2: Can authors conduct ablation studies on each NEP component?*), so we address them together here.
> >
> > First, we would like to clarify that NEP is introduced as a training task, not as a new architecture or a loss. NEP supervision singal can be instantiated using the standard token-level cross-entropy loss or the GRPO loss, in exactly the same way as QA (you can make GRPO loss or cross-entropy loss over QA data). Thus, there are no special "NEP loss components" to ablate. What we do ablate is precisely how the NEP training signal is defined and used, at three levels:
> >
> > * Task-level comparison under controlled data ($\\textrm{\\color{blue}Tables 1}$ and $\\textrm{\\color{blue}3}$). We fix the backbone architecture, video source, and the amount of video data, and only vary the training task: NEP vs. captioning vs. multiple-choice QA vs. open-ended QA. This isolates the contribution of NEP as a task relative to other standard objectives under the same model and data budget.
> > * Objective instantiation of NEP ($\\textrm{\\color{blue}Table 2}$). We then fix the NEP task itself (same questions, same videos) and compare different ways to instantiate the supervision: SFT, CFT, and Distill. All of these still use cross-entropy, but differ in how the target text is constructed. This ablates how the NEP signal is injected into training.
> > * Temporal coverage of NEP supervision ($\\textrm{\\color{blue}Table 4}$, $\\textrm{\\color{blue}Figure 14}$). Finally, we fix both the NEP questions and answers and vary only the video context provided to the model (partial vs. full video). This isolates the effect of how much temporal evidence is made available to the NEP loss.
> >
> >
> >
> > ---
> >
> > ***Q3: Can authors discuss computational cost and scalability trade-offs of NEP relative to captioning or QA fine-tuning?***
> >
> > Thank you for raising this point. Our comparison focuses on two aspects: computational cost and scalability.
> > * Computational cost: NEP fine-tuning is typically more efficient than standard video QA or captioning. Both video QA and captioning models usually take the full video as input, and the visual tokens dominate the context length and thus the computational cost. In contrast, NEP only consumes the preceding part of the video as input when predicting future events, which substantially reduces the number of tokens for visual information and therefore the overall compute per example, under the same model architecture and training pipeline.
> > * Scalability: On the data side, the construction cost of NEP training data is comparable to that of video QA: both require temporally grounded annotations derived from underlying video captions. This is more expensive than pure captioning, which only needs a single description per clip without additional reasoning structure. However, once such annotations are available, NEP scales similarly to QA, and can be trained within the same infrastructure as standard supervised fine-tuning.
> >
> >
> > ---
> >
> > ***Q4: What are the typical failure modes of the NEP model when predicting future events? How does the model’s performance degrade when event boundaries are noisy or poorly segmented? Could the authors provide a qualitative analysis or visualization of failed predictions to clarify these limitations?***
> >
> >
> > To address the reviewer’s question about the failure modes of the NEP model when predicting future events, we have added a dedicated discussion in $\\textrm{\\color{blue}Appendix C.3}$ along with qualitative examples in $\\textrm{\\color{blue}Figures 14}$ and $\\textrm{\\color{blue}15}$. Contrary to the hypothesis that errors are mainly caused by noisy or poorly segmented event boundaries, we do not observe this as the dominant issue in our analysis. Instead, we find that the model often produces future predictions that are narratively reasonable but only weakly grounded in the specific question description and answer options. We hypothesize that this is partly because NEP-only training trades off some instruction-following ability in exchange for improved temporal understanding and reasoning (as shown in $\\textrm{\\color{blue}Figures 12}$ and $\\textrm{\\color{blue}13}$
> > ). We view this trade-off as expected in our current setup, since we deliberately fine-tune solely on NEP data—without blending in general instruction data—in order to isolate the effect of NEP for study. In turn, these results suggest that, for building a new state-of-the-art system, an appropriate data-blending recipe that combines NEP with broader instruction-following training will be important.

---

> ### Author Response · Authors · 2025-11-26
> **Rebuttal by Authors [3/3]**
>
> ---
>
> ***Q5: How does NEP handle scenarios with multiple plausible futures?***
>
> Thank you for this question. We fully agree that, given a video prefix, the future is inherently multi-modal and many continuations can be reasonable. In NEP, each training instance is defined by one video and its realized continuation: the target text describes what actually happened in the future segment of that particular video. Conceptually, NEP therefore learns a conditional distribution over future events rather than a single deterministic outcome.
>
> Note, this is directly analogous to next-token prediction in language modeling. Given a narrative prefix such as "She stood at the edge of the forest, gripping the letter in her hand as the sun began to set…", there are many continuations that are a priori reasonable (she enters the forest, she returns to the village, someone appears behind her, etc.), but the corpus only contains the single continuation that happened to be written in that story. We do not assume that this observed continuation is the only logically valid one; we treat it as one realized sample from a broader set of plausible futures, and across many such samples the model learns a distribution over next events. *This is exactly how large language models are routinely trained in practice.*
>
> *Our NEP setting follows the same principle*: for a given video prefix, we acknowledge that multiple futures could be reasonable in principle, but for each training example we simply supervise on the specific future that actually occurred in that video. The soundness of NEP does not rely on the future being uniquely determined by the past; it only requires that each example has one coherent realized continuation, which serves as a sample from the underlying distribution of plausible futures.

---

> > ### Comment · Reviewer_m8KR · 2025-11-28
> >
> > Thanks for authors' reply. The new qualitative visualizations are really helpful and make the NEP supervision and model behavior clearer. The added error analysis and failure cases also address my earlier concerns. Authors may talk more about caption analysis for the ablations to strengthen the paper.

---

> ### Author Response · Authors · 2025-12-01
> **Response to the Reviewer’s Follow-Up Comment**
>
> Thank you for your thoughtful follow-up and for revisiting our rebuttal. We are glad that the new qualitative visualizations, error analysis, and failure cases helped clarify how NEP supervision affects the model’s behavior!
>
> We agree that caption analysis is an important component and can further strengthen the paper. In the revised version, we add an ablation on caption analysis and expand the accompanying discussion in $\\textrm{\\color{blue}Section 5.2}$ to more clearly demonstrate its effectiveness. For more details, you may refer to our further discussion with Reviewer ETi3 and the updated Section 5.2 in the revision.

---

### Official Review · Reviewer_gJoZ · 2025-11-03

**Soundness:** 3
**Presentation:** 3
**Contribution:** 3
**Rating:** 4
**Confidence:** 4

**Summary:**

This paper proposes next event prediction, a learning task to foster temporal reasoning. It drives the model to integrate visual perception from the visual encoder with commonsense knowledge in the LLM.
The authors create V1-33K to facilitate tuning. And also they introduce FutureBench to evaluate logical coherence and causal consistency in predicting unseen future events.
The results trained with NEP show competitive performance.

**Strengths:**

1. The motivation of the paper is clear and the paper is well written.
2. Different tuning strategies have been explored and results are reported on several existing benchmarks and the proposed benchmark.
3. It shows improvement on temporal reasoning tasks when training with NEP.

**Weaknesses:**

1. The generated future captions may reflect language priors (“after running, people usually jump”) rather than visual inference, so the causal reasoning claim remains unsubstantiated to me. It would be great if there is analysis shows whether NEP-trained models actually attend to temporal cues or just leverage textual priors.
2. The authors highlighted "SFT remains a simple yet efficient approach for training on NEP". But from Table 2, SFT didn't seem to be the best strategy?
3. The evaluation table seems not including other previous approaches, like VIdeoLlama, LongVA, LLaVA-VIdeo, InternVL, etc.

**Questions:**

1. How do you ensure that QAs in FutureBench needs reasoning from video frames? Do you have a baseline that training with only previous texts and future texts?

---

> ### Author Response · Authors · 2025-11-26
> **Rebuttal by Authors [1/2]**
>
> Thank you for your constructive review and suggestions. We are grateful for the recognition of our paper's "clear motivation" and the improvements on temporal reasoning brought by NEP. Below we respond to the comments in **Weaknesses (W)** and **Questions (Q)**.
>
> ---
>
> ***W1: The generated future captions may reflect language priors (“after running, people usually jump”) rather than visual inference, so the causal reasoning claim remains unsubstantiated to me. It would be great if there is analysis shows whether NEP-trained models actually attend to temporal cues or just leverage textual priors.***
>
>
> Thanks for the good question. We address whether NEP-trained models truly use temporal visual cues rather than only language priors from three views: task design, quantitative results, and qualitative analysis.
>
>
> **(1) Task and training signal (future events, not short-term actions).**
> NEP is defined as predicting future events (at the ***storyline level***) conditioned on the preceding video segment, rather than predicting the next action or frame. As illustrated in the newly added training-data visualizations ($\\textrm{\\color{blue}Figures 10}$ and $\\textrm{\\color{blue}11}$), each training sample pairs a video segment with a textual description of how the future event will unfold. This goes beyond short-range patterns "after running, people usually jump" (*next action prediction*) which can be guessed from language priors alone. In fact, compared to *next-frame* or *next-action* prediction used in many related works [1,2,3], a key advantage of our next-event prediction formulation is that it explicitly encourages the model to jointly use (i) temporally grounded visual evidence about how the scene is evolving and (ii) commonsense priors about how multi-step events typically unfold, rather than relying on either source in isolation. The training data therefore explicitly encode a visual, temporally reasoning signal over multi-step event progressions.
>
>
> **(2) Quantitative evidence on temporal benchmarks (beyond textual priors).**
> If NEP were primarily exploiting textual priors, we would expect limited gains on benchmarks where simple language heuristics are insufficient. However, as shown in $\\textrm{\\color{blue}Table 1}$ and the newly added $\\textrm{\\color{blue}Table 3}$ (*on the advanced temporal reasoning benchmark TOMATO*), NEP consistently improves temporal reasoning performance over the same base model. On TOMATO, which is designed to stress temporal ordering and coherence, NEP yields a +2.42 point improvement over the baseline.
>
> Moreover, on our FutureBench benchmark, we explicitly test whether strong language priors alone are sufficient by converting all videos into textual descriptions and evaluating a powerful text-only reasoning model (o4-mini) without any visual input. This model achieves only 32.0% accuracy, indicating that even advanced text-only reasoning struggles to solve these tasks from language alone. In contrast, our NEP-tuned video model attains higher accuracy under the same evaluation protocol.
>
>
> Those gains appear specifically on tasks where the model must respect visual temporal structure, supporting our claim that NEP improves the integration of temporal cues rather than merely amplifying language priors.
>
>
> **(3) Qualitative analysis: attention to temporal structure vs. priors.**
> To directly examine how models use temporal information, we added case studies comparing Qwen2.5-VL-7B with its NEP-trained variant ($\\textrm{\\color{blue}Figures 12}$ and $\\textrm{\\color{blue}13}$ and $\\textrm{\\color{blue}Appendix C.3}$). In the laundry-room example ($\\textrm{\\color{blue}Figure 12}$), the NEP-trained model correctly treats "taking the mug and leaving" as events will happen and selects the option that continues the ongoing laundry process as the future event. In contrast, the vanilla model fixates on the mug, effectively treating "grab mug and leave" as the main future event and rejecting the correct laundry continuation, which reflect the NEP-trained model has a better understanding of the temporal order. In the dog-grooming example ($\\textrm{\\color{blue}Figure 13}$), the base model’s prediction aligns with a generic prior of "more grooming scenes" whereas the NEP-trained model picks intermediate product-display shots that better match the actual ad-like chronology leading up to the final exterior storefront. Here, the NEP model’s choice is driven by the video structure (store interior -> products -> final exterior sign), not by a simple textual association like "grooming -> more grooming".
>
>
> Together, the task definition, the quantitative improvements on temporally demanding benchmarks, and these qualitative examples indicate that NEP-trained models are indeed leveraging temporal visual cues and event progression, rather than merely amplifying language-only priors.

---

> > ### Author Response · Authors · 2025-11-26
> > **Rebuttal by Authors [2/2]**
> >
> > ***W2: The authors highlighted "SFT remains a simple yet efficient approach for training on NEP". But from Table 2, SFT didn't seem to be the best strategy?***
> >
> > Thank you for the question and for pointing out this potential ambiguity. In the current draft, our statement that "SFT remains a simple yet efficient approach for training on NEP" uses SFT in a broad sense: we refer to the whole family of supervised fine-tuning objectives that train with a cross-entropy loss on text outputs given the video input. Under this broad notion, all strategies in $\\textrm{\\color{blue}Table 2}$ (including SFT, Distill, and CFT) are SFT-style methods, as they only differ in the exact textual supervision (original next-event descriptions vs. augmented outputs from a teacher LLM), but they share the same supervised loss and training pipeline, in contrast to RL-style objectives. We have clarified this in the revision to avoid the potential confusion.
> >
> >
> > ---
> >
> > ***W3: The evaluation table seems not including other previous approaches, like VIdeoLlama, LongVA, LLaVA-VIdeo, InternVL, etc.***
> >
> > Our experimental design is centered on isolating the training objective rather than proposing a new SoTA video-LLM. Specifically, NEP is introduced as a training task and supervision signal, and in $\\textrm{\\color{blue}Table 1}$ we compare it against other training tasks on the same base model, such as standard captioning, open-ended QA, and close-ended (MC) QA. This controlled setting allows us to attribute performance differences directly to the type of supervision (NEP vs. non-NEP), instead of confounding them with changes in architecture. Including them in the main table would make it harder to draw clear conclusions about the benefit of NEP itself.
> >
> > ---
> >
> >
> > ***Q1: How do you ensure that QAs in FutureBench needs reasoning from video frames? Do you have a baseline that training with only previous texts and future texts?***
> >
> >
> > Thanks for the question. We address this point in $\\textrm{\\color{blue}Section 4.2}$; here we summarize and make it more explicit.
> >
> > First, when constructing FutureBench QA pairs, we deliberately design the questions so that the correct option cannot be determined from language priors or local lexical cues alone: (i) the question only describes the high-level situation without “leaking” the answer, (ii) the key discriminative signals are visual and (iii) distractor options are written to be linguistically plausible but visually inconsistent with the actual video.
> >
> >
> > Second, instead of training a separate text-only NEP model, we adopt a strong text-only reasoning baseline that uses only previous and future textual descriptions. Concretely, we use the textual captions of videos and evaluate the o4-mini model without any visual input. This text-only baseline achieves 32.0% accuracy, indicating that even advanced language-only reasoning is insufficient to consistently solve FutureBench. In contrast, video models perform substantially better (as shown in $\\textrm{\\color{blue}Table 1}$), supporting that FutureBench truly requires video frames rather than relying solely on language priors.
> >
> >
> > ---
> >
> > ***Referencs***
> >
> >
> > [1] Liu, Xin, et al. "From recognition to prediction: Leveraging sequence reasoning for action anticipation." ACM Transactions on Multimedia Computing, Communications and Applications 20.11 (2024): 1-19.
> >
> > [2] Beedu, Apoorva, et al. "On the efficacy of text-based input modalities for action anticipation." arXiv preprint arXiv:2401.12972 (2024).
> >
> > [3] Assran, Mido, et al. "V-jepa 2: Self-supervised video models enable understanding, prediction and planning." arXiv preprint arXiv:2506.09985 (2025).

---

> > > ### Comment · Reviewer_gJoZ · 2025-11-28
> > >
> > > Thank you for the response. Most of my concerns have been addressed, and I will consider raising my score. I have also read other reviews, and I some  additional questions. Regarding Table 2, following my earlier comment, could you clarify the mixed tuning strategy in more detail? Like, how does the model integrate the different supervision signals within each training epoch? Do you partition the data according to different supervision types, or is everything mixed within the same batches? L423 states that the mixed strategy achieves the highest average performance on the temporal benchmarks. However, Table 2 seems to suggest that no single strategy consistently outperforms all others. Could you explain this discrepancy? Finally, how did you determine the choice of 33K samples for tuning? Have you evaluated how tuning data scale affects performance?

---

> > > > ### Author Response · Authors · 2025-11-28
> > > > **Response to Reviewer’s Follow-Up Questions**
> > > >
> > > > We thank the reviewer for the encouraging and timely follow-up, and for carefully revisiting our rebuttal and the other reviews! We address the follow-up questions below.
> > > >
> > > > ---
> > > >
> > > > ***Q1: Following my earlier comment, could you clarify the mixed tuning strategy in more detail? Like, how does the model integrate the different supervision signals within each training epoch? Do you partition the data according to different supervision types, or is everything mixed within the same batches?***
> > > >
> > > >
> > > > Thanks for your question. In the Mix setting, we fix a total tuning budget (e.g., 10K samples) and then allocate this budget equally across supervision types. Concretely, we sample the same number of examples from each supervision type (e.g., for SFT, Distill, and CFT, sample roughly 10K/3 examples from each), concatenate them into a single mixed dataset, and then shuffle this unified pool at the beginning of each epoch.  During training, mini-batches are sampled from this unified mixed pool.
> > > >
> > > > This clarification makes the description of the mixed strategy more precise, and we have added it in the $\\textrm{\\color{blue}Section 5.2}$.
> > > >
> > > >
> > > > ---
> > > >
> > > >
> > > >
> > > > ***Q2: L423 states that the mixed strategy achieves the highest average performance on the temporal benchmarks. However, Table 2 seems to suggest that no single strategy consistently outperforms all others. Could you explain this discrepancy?***
> > > >
> > > > As shown in the rightmost column (T-Avg.) of Table 2, the mixed strategy achieves the average score of 53.3 (+3.6), which is the highest among all strategies.
> > > >
> > > >
> > > > ---
> > > >
> > > >
> > > > ***Q3: How did you determine the choice of 33K samples for tuning? Have you evaluated how tuning data scale affects performance?***
> > > >
> > > >
> > > > Thank you for this question. Our choice of 33K tuning samples is driven by the goal of making a fair comparison between NEP and existing training tasks. Concretely, we start from the same underlying video corpus that has been used in prior video instruction-tuning works, e.g., the data used for *both* captioning and QA style instruction tuning. Since NEP is a new training task, we want to compare it against conventional training objectives (captioning, QA, etc.) under as similar data conditions as possible (e.g., source and amount). From this shared corpus, we then randomly sample candidate clips and construct 33K NEP instances.
> > > >
> > > >
> > > > Regarding data scale, we agree that systematically studying scaling laws for NEP (e.g., varying from 10K to 100K+ samples) is an interesting and important direction, but it is beyond the computational budget of the current work, so we focus on this fixed-budget setting to study the effect of the proposed NEP training task in a controlled manner. We also expect that performance is influenced not only by the quantity of NEP data but also by the diversity of video sources (different domains, motion patterns, and event types), which could further benefit temporal reasoning. A more thorough exploration of both data scale and data diversity for NEP is a promising direction for future work, and we will clarify this in the revised version.

---

### Author Response · Authors · 2025-12-03
**Summary of Rebuttal Revisions and General Response**

Dear Reviewers and Area Chairs,

Thank you for taking the time to review our paper!
We are glad that the reviewers recognize the strengths of our Next-Event Prediction (NEP) task in improving temporal reasoning. After our initial rebuttal, all four reviewers revisited our responses and provided timely and supportive follow-up feedback. In response, we added new empirical results and clarifications. We are also encouraged that one reviewer (gJoZ) explicitly mentioned that they are considering raising their score.

Below we summarize the main concerns and what we addressed in the rebuttal and revision:

| Reviewer | Main concerns (very brief) | What we did in rebuttal / revision |
| :------- | :------------------------- | :--------------------------------------------- |
| **gJoZ** | Temporal cues vs. language priors; wording around “SFT” vs. Table 2; FutureBench’s reliance on video vs. text-only; clarity on mixed tuning and 33K data. | Clarified that NEP is defined at **storyline-level future events** (not next-frame/action) and added **training-data visualizations** for NEP samples (new Figs. Appx C.2) to show temporally grounded supervision; reported **advanced temporal benchmark results**, including **TOMATO** (new Table 4), and the **text-only o4-mini baseline** on FutureBench showing that language-only reasoning is insufficient; added **qualitative case studies** (laundry-room, dog-grooming; Appx C.3) contrasting NEP vs. Qwen2.5-VL-7B; clarified that all strategies in Table 2 are SFT-style cross-entropy objectives differing only in supervision text (SFT / Distill / CFT); and detailed the **mixed-tuning implementation** (fixed 10K budget, equal allocation per supervision type) and the rationale for **33K NEP samples** plus a discussion of scaling and diversity. |
| **m8KR** | Missing qualitative visualizations and error/failure analysis; unclear isolation of NEP vs. other objectives; limited discussion of compute / scalability and multi-future nature. | Added **visualizations of NEP training samples** (Appx C.2) and **case studies of predicted futures** comparing Qwen2.5-VL-7B vs. NEP-tuned models; extended **success and failure analyses** to highlight typical error patterns (Appx C.3); reorganized experiments into three ablation axes: (i) **task-level comparison** vs. caption/OEQA/MCQA under shared data (Tables 1, 4), (ii) **objective instantiations** of NEP (SFT / Distill / CFT, Table 2), and (iii) **temporal coverage** of visual context (Table 5); and discussion on **compute cost vs. caption/QA** and **data-side scalability**, as well as how NEP handles **multiple plausible futures** via a conditional distribution over realized continuations. |
| **ETi3** | Data-scaling saturation at 5K and limited dataset diversity; no empirical evidence that Caption Analysis helps; FutureBench quality not fully justified. | Added an extended **scaling study** (Sec. 5.2) and a four-factor explanation of saturation (NEP’s temporal focus, restricted intersection of video sources, NEP-only tuning under QA-style evaluation, and limitations of existing temporal benchmarks), and showed that on **TOMATO** NEP exhibits a more positive scaling trend (Fig. 6); ran a new **Caption Analysis ablation** (Table 3) comparing NEP tuning **with vs. without** Caption Analysis on the same corpus, demonstrating consistent temporal gains and clarifying the “optional” wording in Sec. 3.2; and strengthened **FutureBench quality evidence** by emphasizing our **human-in-the-loop review** (Appx. B.3) and adding **FutureBench examples** (Figs. 8–10, 13–14). |
| **YKsb** | NEP curation under ambiguous futures; GRPO only on FutureBench with general-performance drop; request for more temporal benchmarks (e.g., TOMATO); lack of FutureBench statistics. | Clarified the **soundness of NEP curation** via an explicit analogy to **next-token LM training**: each video prefix is paired with its realized future as one sample from a broader set of plausible continuations, and the judge is designed to tolerate minor paraphrases, so correctness does not require a unique deterministic future; explained that **GRPO on NEP** is kept as a secondary configuration that boosts FutureBench but slightly hurts general benchmarks, reinforcing **SFT as the robust default**; added **TOMATO evaluation** (Table 4) showing NEP-driven gains on a harder temporal benchmark; **FutureBench dataset statistics** (Table 7) on video length and difficulty distributions. |

In follow-up replies, reviewers (gJoZ, m8KR, YKsb) generally acknowledged that these additional experiments and clarifications improved the completeness and presentation of the work.

We understand that this year’s ICLR review process has been especially challenging, and we are sincerely grateful that you took the time to read our lengthy rebuttal and additional experiments. We remain confident that the Next-Event Prediction (NEP) as a learning task can meaningfully contribute to improve VLM temporal reasoning!

---

### Meta-Review · Area_Chair_WwCJ · 2026-01-07

**Summary:**

This paper received scores of 4,4,6,4.  Initial concerns include causal reasoning claim being unsubstantiated, some unclear/ambiguous result claims, missing baselines, lacking qualitative visualization of predicted events or linguistic coherence, lacking error analysis or failure case study contrasting NEP and baseline models, lacking ablation on architecture or training objectives to isolate contribution of NEP loss components, dataset lacking sufficient diversity or scale, dataset may contain samples that are either trivial or excessively difficult, insufficient evidence supporting the quality of FutureBench samples, suggestion to evaluate on other temporal benchmarks (e.g., Vinoground, Tomato), and GRPO performance only reported on FB but not other temporal benchmarks.

Overall, the reviewers find the core idea to be sound and innovative, the paper well-written, and other positive points include good results on extensive experiments and a novel benchmark.  The ACs view the strengths to outweigh the weaknesses, and recommend acceptance.

**Reviewer Concerns:**

The major concerns were adequately addressed by the rebuttal.

**Reviewer Scores:**

Post-rebuttal, the reviewers may have converged around 6.

---

### Decision · Program_Chairs · 2026-01-26

Accept (Poster)